# Functionalized graphene-oxide grids enable high-resolution cryo-EM structures of the SNF2h-nucleosome complex without crosslinking

Un Seng Chio [1,9], Eugene Palovcak[2,9], Anton A. A. Smith[3,8], Henriette Autzen[1,4], Elise N. Muñoz[5], Zanlin Yu [1], Feng Wang[1], David A. Agard [1], Jean-Paul Armache [6] ✉, Geeta J. Narlikar [1] ✉ & Yifan Cheng [1,7] ✉

Single-particle cryo-EM is widely used to determine enzyme-nucleosome complex structures. However, cryo-EM sample preparation remains challenging and inconsistent due to complex denaturation at the air-water interface (AWI). Here, to address this issue, we develop graphene-oxide-coated EM grids functionalized with either single-stranded DNA (ssDNA) or thiol-poly(acrylic acid-co-styrene) (TAASTY) co-polymer. These grids protect complexes between the chromatin remodeler SNF2h and nucleosomes from the AWI and facilitate collection of high-quality micrographs of intact SNF2h-nucleosome complexes in the absence of crosslinking. The data yields maps ranging from 2.3 to 3 Å in resolution. 3D variability analysis reveals nucleotide-state linked conformational changes in SNF2h bound to a nucleosome. In addition, the analysis provides structural evidence for asymmetric coordination between two SNF2h protomers acting on the same nucleosome. We envision these grids will enable similar detailed structural analyses for other enzyme-nucleosome complexes and possibly other protein-nucleic acid complexes in general.

Since the first structure of the nucleosome was determined 25 years ago[1], considerable efforts have been made to understand how different chromatin factors engage nucleosomes. Determining X-ray crystal structures of nucleosome-protein complexes proved to be challenging, and fewer than ten such structures have been reported to date[2–4]. However, after the resolution revolution in single-particle cryogenic electron microscopy (cryo-EM) in 2013[5], this method has now been used to determine over 40 different nucleosome-protein complex structures[3,4]. Thus, single-particle cryo-EM has become the method of choice for elucidating structures of nucleosome-protein complexes.

Despite this newfound success, it remains a non-trivial task to determine nucleosome-protein complex structures by cryo-EM. A major bottleneck is the preparation of suitable cryo-EM grids of these complexes for data collection. Nucleosome-protein complexes tend to fall apart upon plunge freezing presumably due to particle denaturation at the air-water interface (AWI). Indeed, many cryo-EM structures

[1]Department of Biochemistry and Biophysics, University of California San Francisco, San Francisco, CA, USA. [2]Biophysics Graduate Program, University of California San Francisco, San Francisco, CA, USA. [3]Department of Materials Science & Engineering, Stanford University, Stanford, CA, USA. [4]Linderstrom-Lang Centre for Protein Science, Department of Biology, University of Copenhagen, København, Denmark. [5]Tetrad Graduate Program, University of California, San Francisco, San Francisco, CA, USA. [6]Department of Biochemistry and Molecular Biology and the Huck Institutes of the Life Sciences, Pennsylvania State University, University Park, PA, USA. [7]Howard Hughes Medical Institute, University of California San Francisco, San Francisco, CA, USA. [8]Present address: Department of Health Technology, Technical University of Denmark, Kongens Lyngby, Denmark. [9]These authors contributed equally: Un Seng Chio, Eugene Palovcak. ✉e-mail: jparmache@psu.edu; geeta.narlikar@ucsf.edu; yifan.cheng@ucsf.edu

of nucleosome-protein complexes were determined from samples that were chemically crosslinked prior to conventional plunge freezing[6–13]. However, it remains unclear whether the use of crosslinkers may introduce artifacts and/or limit conformational diversity. Without crosslinking, preparation of sample grids is highly variable. From our own experience working with a nucleosome bound with the ATP-dependent chromatin remodeler SNF2h, a few suitable grids with intact complexes were obtained only by chance after many trials[14]. This high variability also demands large amounts of protein for preparing a large number of sample grids. While we can generate sufficient recombinant SNF2h from *E. coli*, it is challenging to obtain large amounts of other chromatin factors directly purified from an endogenous source. Thus, improving our ability to generate high-quality cryo-EM grids of intact complexes in a reproducible manner would greatly facilitate determining structures of a wider range of nucleosome-protein complexes.

To address these issues, other groups have developed methods to prevent nucleosome-protein complexes from contacting the AWI[15–21]. However, these methods either require the use of sophisticated equipment and/or materials for grid preparation. Here we present a straightforward and reproducible method for determining structures of the SNF2h-nucleosome complex using graphene-oxide (GO) grids functionalized with negatively charged DNA or synthetic polymers. We chose to use SNF2h as a test case because SNF2h adopts both monomeric and dimeric states on a nucleosome, with the dimeric state being more active[22–24]. However, past attempts to obtain high-resolution structures of the dimeric state have been limited[14].

In this work, we build upon a previously established simple procedure for layering GO on top of EM grids without the need for sophisticated equipment[25]. Functionalization of these GO grids with single-stranded DNA (ssDNA) or an amphiphilic polymer with comparable structure and charge properties leads to stabilization of nucleosome-protein complexes for single-particle cryo-EM (Fig. 1a). The method is highly reproducible and requires 10-fold less sample than with normal Quantifoil holey carbon grids. Using this method, we obtain high-quality single-particle cryo-EM datasets and determine high-resolution structures of the SNF2h-nucleosome complex. By

further classifying particles within the dataset, we identify additional monomeric and dimeric SNF2h states bound to a nucleosome that deepen our understanding of SNF2h mechanism. We envision that these functionalized GO grids will be applicable for other samples involving nucleic acid-protein complexes, as well as other samples with charged surfaces.

## Results

### Functionalized GO grids for single-particle cryo-EM of chromatin samples

We previously determined cryo-EM structures of the ATP-dependent chromatin remodeler SNF2h bound to nucleosomes at both the superhelical location (SHL)−2 and SHL+2 positions in the presence of the non-hydrolysable ATP analog ADP-BeF$_x$[14]. In our hands, without crosslinking, SNF2h frequently dissociates from the nucleosome upon plunge freezing. Although we determined a 3.4 Å structure of a single SNF2h bound to nucleosome at the SHL-2 position, dissociation of the complex hindered routine structural studies by cryo-EM, hindering detailed mechanistic studies to understand how SNF2h translocates DNA around a nucleosome. This early work also demonstrated that the AWI is a major factor causing the complex dissociation. Therefore, we sought to improve our ability to generate reproducible cryo-EM samples of the SNF2h-nucleosome complex.

To protect chromatin complexes from damage at the AWI during cryo-EM grid preparation, our initial idea was to functionalize GO-coated EM grids with ssDNA that is complementary in sequence to an ssDNA overhang engineered onto the flanking DNA of nucleosomes (Supplementary Fig. 1a). We envisioned that the complementary sequences would anneal, thus allowing the functionalized GO surface to effectively capture chromatin complexes and keep them away from the AWI during plunge freezing. We functionalized GO-coated grids using a 15-base oligonucleotide with a primary amine at the 5′ end (see Methods) that can undergo nucleophilic reaction with epoxide groups on the GO surface in nonaqueous conditions[26,27]. These ssDNA GO grids allowed us to prepare cryo-EM grids of the intact SNF2h-nucleosome complex successfully and reproducibly. Subsequently, we discovered that sequence complementarity is not necessary, as ssDNA

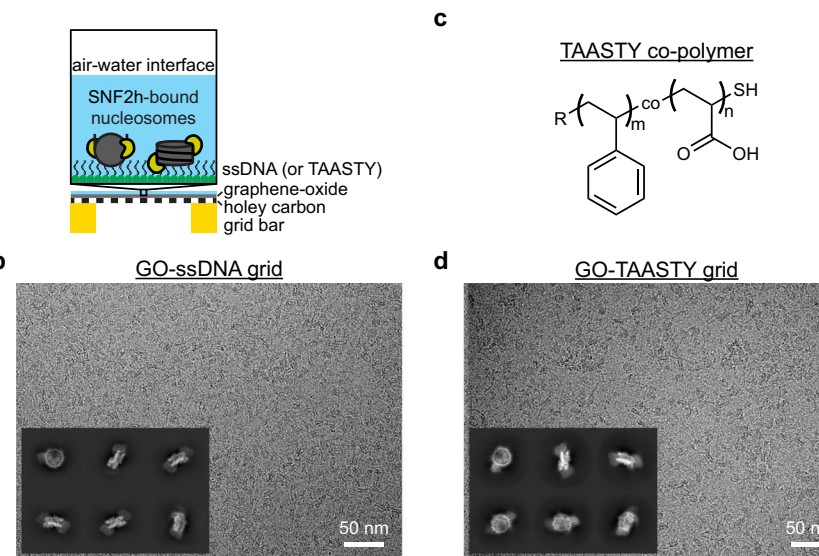

**Fig. 1 | Functionalized GO grids protect SNF2h-nucleosome complexes from the air-water interface.** **a** Schematic illustrating functionalized graphene-oxide (GO) grids. GO is layered on top of a commercial Quantifoil holey carbon grid. The GO surface is then functionalized with either single-strand DNA (ssDNA) or thiol-poly(acrylic acid-co-styrene) (TAASTY) co-polymer to attract SNF2h-nucleosome complexes away from the air-water interface. **b** Representative micrograph out of 7423 micrographs of the SNF2h-nucleosome complex on a ssDNA GO grid. Representative 2D classes of particles picked from a ssDNA GO grid are also shown. **c** Chemical structure of the TAASTY co-polymer. **d** Representative micrograph out of 2,020 micrographs of the SNF2h-nucleosome complex on a TAASTY GO grid. Representative 2D classes of particles picked from a TAASTY GO grid are also shown.

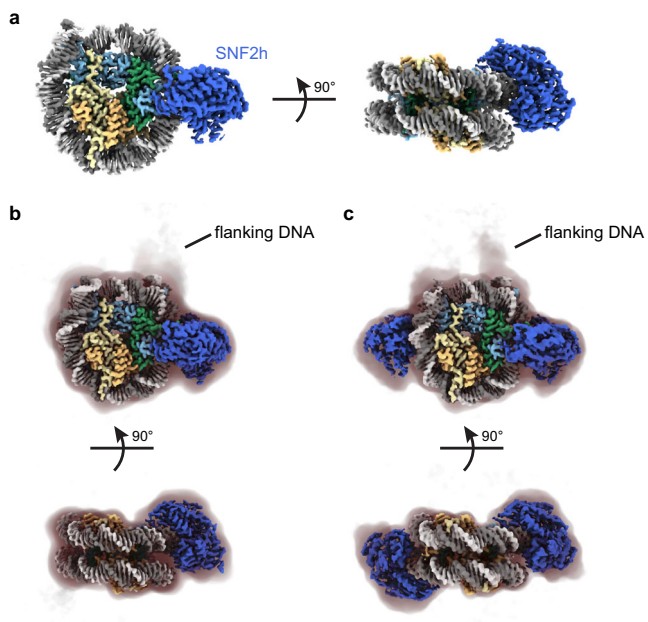

**Fig. 2 | High-resolution SNF2h-nucleosome structures determined using functionalized GO grids. a** Coulomb potential map of a consensus SNF2h-nucleosome complex at ~2.3 Å global resolution determined with datasets collected on ssDNA GO and TAASTY GO grids. In all figures, histones H3, H4, H2A, and H2B are colored as light blue, green, yellow, and orange, respectively. SNF2h is colored in darker blue. The two strands of DNA are colored as light and dark gray. **b** Coulomb potential map of SNF2h bound to nucleosome at the inactive position at ~2.7 Å resolution. A filtered map at lower contour is shown as a shadow with the position of flanking DNA marked. **c** Coulomb potential map of the double-bound SNF2h-nucleosome complex at ~2.8 Å resolution. A filtered map at lower contour is shown as a shadow with the position of flanking DNA marked.

GO grids enabled preparation of cryo-EM grids of SNF2h-nucleosome complex even without the complementary single-strand overhang on the nucleosome (Fig. 1b). Thus, we speculated that the negatively charged ssDNA is able to attract nucleosomes to the GO surface via nonspecific electrostatic interactions. To mimic the negative charge of ssDNA, we also made functionalized GO grids with short thiol-poly(acrylic acid-co-styrene) (TAASTY) co-polymers, which have similar charge properties and chemical structure to ssDNA (Fig. 1c). Indeed, we found that the TAASTY GO grids are able to facilitate the preparation of cryo-EM grids of chromatin complexes (Fig. 1d).

Using the ssDNA and TAASTY GO grids, we were able to reproducibly prepare cryo-EM grids of SNF2h-nucleosome complexes (Fig. 1b, d), allowing routine collection of large single-particle cryo-EM datasets of intact SNF2h-nucleosome complex (Fig. 1b, d, insets), suggesting that the ssDNA and TAASTY GO grids are able to protect SNF2h-nucleosome complexes from the AWI. The SNF2h-nucleosome complexes are likely retained at the GO surface and away from the AWI via nonspecific electrostatic interactions with the ssDNA and TAASTY co-polymer. This was not possible with regular Quantifoil grids, as repeatedly shown in our previous study, in which SNF2h-nucleosome complexes fall apart on cryo-EM grids despite optimized biochemistry as confirmed by negative-stain EM[14].

Similarly, we were also able to prepare cryo-EM grids of isolated nucleosomes under physiological buffer conditions (Supplementary Figs. 1b, c). Nucleosomes in buffer without salts are stable in cryo-EM grid preparation[28–30], but often fall apart in buffers containing salt, such as 150 mM NaCl, which is often required when forming complexes with other chromatin remodelers. With ssDNA GO grids, we see a dense coating of intact nucleosomes under cryo-EM conditions with the application of only 3 μL of ~100 nM non-crosslinked sample on the

grid (Supplementary Figs. 1b, c), which is normally only possible at 10-fold higher nucleosome concentration with standard Quantifoil grids at low salt concentrations[14,28]. 2D-class averages of the nucleosome particles reveal top and side views (Supplementary Fig. 1d) consistent with previously determined nucleosome structures[28–30]. Thus, ssDNA GO grids effectively protect nucleosome particles from denaturation at the AWI while requiring much lower amounts of sample versus use of commercial Quantifoil grids.

## High resolution structures of the SNF2h-nucleosome complex without crosslinking

As the main target of this study, we focused our efforts on the analysis and determination of high-resolution structures for the SNF2h-nucleosome complex. We generated complexes by mixing 500 nM SNF2h and 100 nM nucleosomes with a 60-base pair DNA overhang on one end in the presence of 60 mM KCl and 2 mM ADP-BeF$_x$ and directly applied 3 μL to ssDNA GO and/or TAASTY GO grids. We confirmed that particles on both types of grids are attracted to the functionalized GO surface away from the air-water interface using electron cryotomography (Supplementary Movies 1 and 2). From datasets collected on both types of grids, we determined a ~2.3 Å global resolution consensus structure of SNF2h bound to a nucleosome (Fig. 2a; Supplementary Fig. 2). This consensus structure consists of both single- and double-bound SNF2h-nucleosome particles. In addition, due to a pseudo 2-fold symmetry of the nucleosome, particles corresponding to SNF2h bound to nucleosomes at the SHL-2 and SHL+2 positions may be averaged together. We therefore performed further focused classification to first separate single- versus double-bound particles (Supplementary Fig. 2). With the single-bound particles, we applied symmetry expansion to identify the location of flanking DNA (Supplementary Fig. 3a), resulting in two maps that correspond to SNF2h bound to nucleosome at either the SHL-2 or SHL+2 positions. Because the resolution of certain DNA base pairs is sufficiently high, purines and pyrimidines can be distinguished from one another, confirming the orientation of the Widom 601 sequence of the nucleosome. This revealed that the SHL + 2 map also in fact corresponded to SNF2h at SHL-2 (Supplementary Figs. 3b, c). We therefore refined all the single-bound particles together, resulting in a ~2.5 Å global resolution map of SNF2h bound to a nucleosome at SHL-2 (Fig. 2b; Supplementary Figs. 3d–f). Inspection of the DNA base pairs confirmed that a majority of the single-bound particles correspond to SNF2h at SHL-2 (Supplementary Fig. 3g). The lack of particles corresponding to single-bound SNF2h at the SHL+2 position is consistent with previous observations at 140 mM KCl with normal Quantifoil grids and is likely due to increased dynamics of the active protomer at SHL+2[14]. While we previously observed more particles bound at SHL+2 at 70 mM KCl with normal Quantifoil grids[14], the ~10-fold higher concentrations of SNF2h and nucleosome in those experiments may have enabled more SNF2h to remain bound at SHL+2 during the plunge freezing process.

For the double-bound particles, we first applied symmetry expansion to identify the flanking DNA (Supplementary Fig. 4a). While most particles appeared to align correctly based on appearance of flanking DNA at low contour levels, 3D variability analysis (3DVA) in cryoSPARC[31] revealed a small population of misaligned particles. Therefore, we applied a few rounds of 3DVA to remove ambiguous particles and retain particles correctly aligned for the flanking DNA; refinement of these particles resulted in a ~2.8 Å global resolution map of SNF2h bound to a nucleosome both at SHL-2 and SHL+2 (Fig. 2c; Supplementary Figs. 4a–c). We see weaker density for SNF2h at the SHL+2 position versus the SHL-2 position (Supplementary Fig. 4d), which is consistent with previous data[14] and with SNF2h at the SHL+2 position being the active protomer and more conformationally flexible[23,32–35].

In contrast to our previously determined structure of the SNF2h-nucleosome complex where we only observed density for ADP

(Supplementary Fig. 5a)[14], we see clear density of intact ADP-BeF$_x$ in the current single-bound structure (Supplementary Fig. 5b). However, we can only see clear density for ADP for both SNF2h protomers in the double-bound structure (Supplementary Fig. 5c). In addition, while we observed a two base pair translocation for SNF2h-bound nucleosome at 140 mM KCl[14], we do not see such translocation in the structures determined here at 60 mM KCl (Supplementary Fig. 6a). This is consistent with previous data collected at 70 mM KCl and corresponding FRET data[14]. We did not notice any significant conformational changes within the nucleosome between the current single-bound and double-bound SNF2h-nucleosome structures (Supplementary Fig. 6b). The RMSD for a majority of histone residues and DNA base pairs were below 1 Å between the two structures with only the flexible flanking DNA exhibiting larger changes, as well as the histone H4 tail, DNA at SHL+2, and DNA at SHL-6 exhibiting larger changes from direct engagement with the second SNF2h protomer.

## SNF2h-nucleosome motions observed at high-resolution

Given the high quality of the current SNF2h-nucleosome dataset, we considered whether we could see either nucleosome and/or SNF2h motions using 3DVA in cryoSPARC. In order for observations to be meaningful, we applied 3DVA to each of the subsets of particles corresponding to either a single-bound SNF2h at SHL-2 or double-bound SNF2h.

For the ~920,000 particles corresponding to nucleosomes with SNF2h at the SHL-2 position, 3DVA revealed six significant variability components. For each component, we isolated two subsets of particles corresponding to each end of the reaction coordinate and computed two new reconstructions (Fig. 3a; Supplementary Figs. 7–9). Therefore, these twelve reconstructions ranging from ~2.8 Å to ~2.9 Å global resolution represent discrete conformations derived from particles within the dataset. We were able to see various global motions in each variability component. In one of the six components, we see a slight squeezing/expansion of the nucleosome in one direction (Supplementary Figs. 7a, b), and in another component, we see squeezing/expansion in a perpendicular direction (Supplementary Figs. 7c, d). The magnitude of squeezing/expansion is ~2 Å, which is less than previously observed in experiments with unbound core nucleosomes[30]. Therefore, SNF2h binding may limit the intrinsic squeezing/expansion fluctuations of a nucleosome.

Several variability components displayed unwrapping of DNA from the nucleosome (Supplementary Fig. 8). In two components, we see DNA unwrapping on the flanking DNA side of the nucleosome (Supplementary Fig. 8a), while in a third component, we see DNA unwrapping on the opposite side of the nucleosome (Supplementary Fig. 8b). Consistent with previous observations[29,36], in two of these components unwrapping of the DNA results in concomitant loss of density for the adjacent histones H3 and H2A. However, we also observe a component where DNA unwrapping does not lead to loss of H3 and H2A density (Supplementary Fig. 8a), suggesting the two phenomena, DNA unwrapping and loss of H3 and H2A density, may not be coupled. Notably, while the CHD family of chromatin remodelers can unwrap DNA from nucleosomes, it remains unclear whether ISWI family remodelers such as SNF2h induce DNA unwrapping[37,38]. Our observations of DNA unwrapping from both the entry and the exit sides suggests that these are stochastic fluctuations and SNF2h does not specifically induce DNA unwrapping unlike CHD remodelers.

In one of the six components, we see rocking of SNF2h on the nucleosome (Fig. 3a; Supplementary Figs. 9 and 10), where both lobes of the SNF2h ATPase shift upwards away from SHL6 and towards the histone octamer core. Dissociation of SNF2h from SHL6 leads to overall weaker density for SNF2h relative to SNF2h density when bound, which suggests that interactions with SHL6 help promote conformational stability for SNF2h on nucleosomes. Previous cryo-EM structures of ISW1, the yeast homolog of SNF2h, bound to nucleosome showed that the ADP-bound state has the top lobe of the ATPase domain dramatically shifted upwards but the bottom lobe similarly positioned away from SHL6 as in the current SHL6-dissociated structure (Supplementary Fig. 9d)[39]. Since our SNF2h sample is prepared with ADP-BeF$_x$, which mimics ATP in a pre-hydrolysis state or activated ATP state, our current structure with SNF2h dissociated from SHL6 likely represents an intermediate between the ATP-bound and post-hydrolysis ADP-bound states. In agreement with this hypothesis, closer inspection of the nucleotide state for each endpoint structure from this variability component shows that SNF2h bound to SHL6 has clear density for ADP-BeF$_x$, whereas SNF2h dissociated from SHL6 does not, and only ADP is clearly visible (Fig. 3b). To test the functional importance of SHL6 contacts, we generated a SNF2h mutant, which we call SNF2h$^{SHL6\_ALA}$, where the entire SHL6 interaction interface is mutated to alanines (from $^{292}$KEKSVFKK$^{299}$ to $^{292}$AEAAVFAA$^{299}$). Under single-turnover conditions with respect to ATP (i.e. SNF2h with trace ATP but with excess and saturating nucleosomes over SNF2h), we observed a ~14-fold defect in ATP hydrolysis by SNF2h$^{SHL6\_ALA}$ compared to wild-type SNF2h (Fig. 3c and Supplementary Fig. 11). Therefore, SNF2h contacts with SHL6 play an important role in facilitating ATP hydrolysis, which is consistent with previous observations where mutating the SHL6 interaction interface in the homolog Snf2 also resulted in decreased ATP hydrolysis activity[40]. Taking the structural and biochemical data together, we propose a speculative model for the SHL6-dependent conformational cycle of SNF2h as it hydrolyzes ATP for DNA translocation that is described in the discussion section below (Fig. 3d).

## Coordinated asymmetric action of SNF2h protomers on a nucleosome

Substantial prior work has indicated that SNF2h functions most optimally as a dimer[22–24]. Yet understanding the structural basis for how the two protomers coordinate their activities has been challenging as the resolution has not been high enough to resolve differences between the protomers[14]. To investigate if our dataset could help distinguish between the two protomers, we applied a similar approach for the double-bound particles and used 3DVA to determine three significant variability components, which resulted in six reconstructions ranging from ~3.1 Å to ~3.2 Å global resolution (Fig. 4a; Supplementary Fig. 12). In one of the variability components, we see squeezing/expansion of the nucleosome similar to observations with single-bound particles (Supplementary Fig. 12b). In the other two variability components, we see the densities for the two SNF2h protomers alternate in strength, where one endpoint has strong density for SNF2h at SHL-2 but weaker density for SNF2h at SHL+2, and vice versa for the other endpoint (Fig. 4a; Supplementary Fig. 12a). Since all the particles in this analysis are double-bound by earlier classification (Supplementary Figs. 2 and 4), this variability cannot be due to differences in SNF2h occupancy at either position. Therefore, these components suggest that there is increased conformational flexibility for one SNF2h protomer on the nucleosome at a given moment.

Although there is no significant conformational change for the SNF2h protomers in one of the two variability components, the second variability component shows a similar SNF2h rocking motion as observed with single-bound particles (Fig. 4a; Fig. 3a). Notably, the two protomers rock asymmetrically on the nucleosome, where while one protomer stably contacts SHL6, the other SNF2h protomer dissociates from SHL6 and becomes more dynamic. The relatively low local resolution of the SNF2h protomers (~5–6 Å) in these maps preclude confident identification of the exact nucleotide state for each protomer. However, considering the proposed model for SNF2h conformational changes during ATP hydrolysis discussed above (Fig. 3d), this asymmetric motion suggests that the two protomers are able to coordinate their actions on a nucleosome such that only one protomer will hydrolyze ATP at a time.

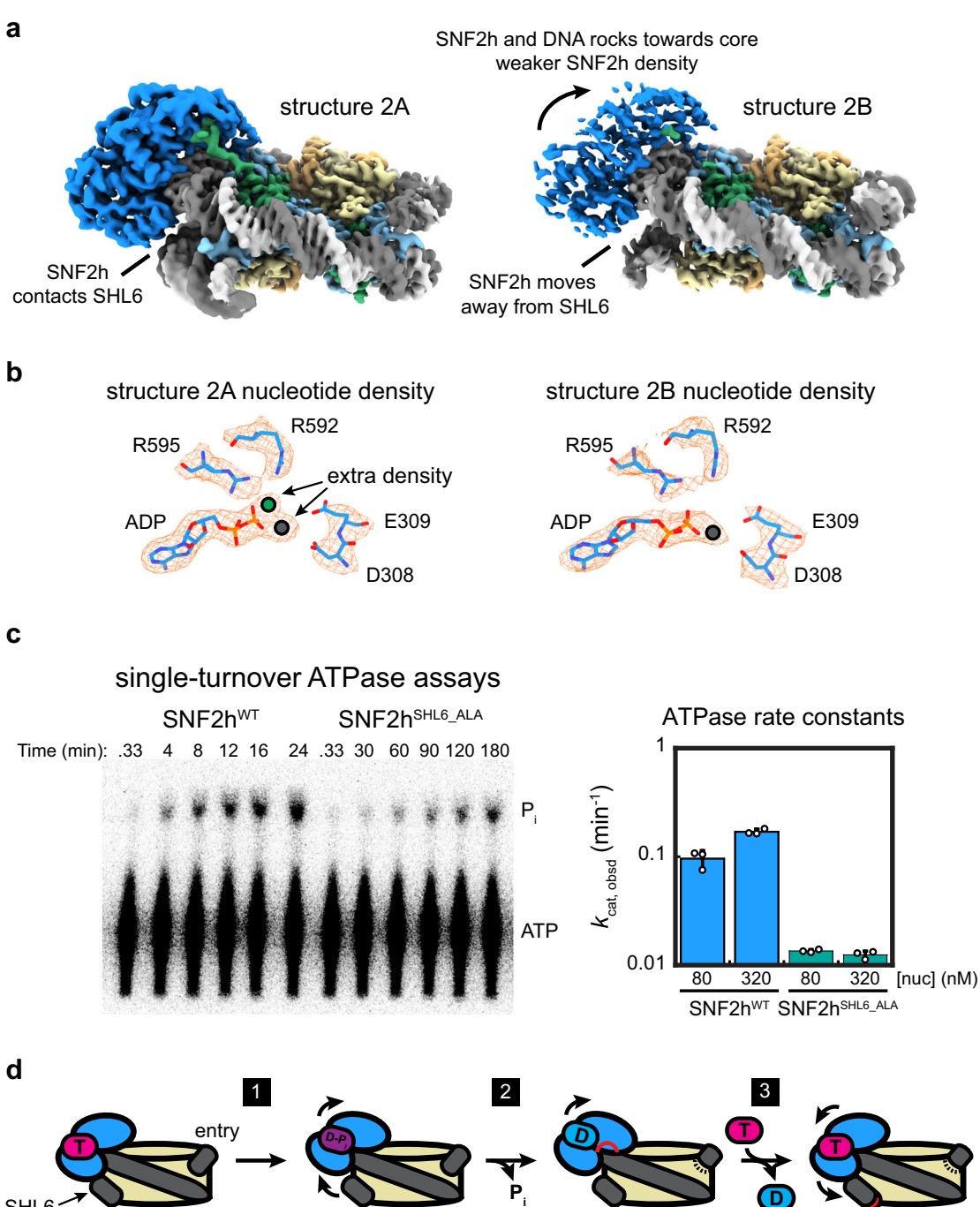

**Fig. 3 | SNF2h rocks on the nucleosome and intermittently contacts SHL6.**
**a** Two coulomb potential maps representing the endpoints of eigenvector 2 from
3D variability analysis of the single-bound SNF2h-nucleosome particles. On one end
(structure 2A), SNF2h is stably bound to nucleosome and contacts DNA at the SHL6
position. On the other end (structure 2B), SNF2h becomes more dynamic and
dissociates from DNA at SHL6 while rocking slightly upward toward the histone
octamer core. **b** Density for nucleotide in structures 2A (threshold level = 0.37) and
2B (threshold level = 0.23). Clear extra density is observed for $Mg^{2+}$ and $BeF_x$ ions in
structure 2A (denoted as gray and green circles, respectively), but in structure 2B
only possible density for $Mg^{2+}$ is observed (gray circle). **c** (left) ATP hydrolysis
assays visualizing hydrolysis using thin-layer chromatography. In the experiments
shown, [nucleosome] = 320 nM, [SNF2h] = 15 nM, and [γ-$^{32}$P-ATP] was in trace.

SNF2h$^{SHL6\_ALA}$ hydrolyzes ATP at a much slower timescale compared to wild-type
SNF2h. Source data are provided as a Source Data file. (right) ATPase rate constants
determined for wild-type SNF2h and SNF2h$^{SHL6\_ALA}$ at [nucleosome] = 80 nM or
320 nM. Data are presented as mean values +/- SD. $n$ = 3 independent ATPase
experiments. Source data are provided as a Source Data file. **d** Model for SNF2h
conformational changes during ATP-dependent translocation of DNA across a
nucleosome. SNF2h is initially in a ground state stably bound to both ATP and DNA
at SHL6. SNF2h then rocks upwards and detaches from DNA at SHL6, entering a
state primed for ATP hydrolysis (step 1). SNF2h hydrolyzes ATP, which causes the
top ATPase lobe of SNF2h to shift upward and promote partial translocation of DNA
(step 2). Exchange of ADP for ATP finishes DNA translocation and resets SNF2h for
subsequent rounds of ATP-dependent remodeling (step 3).

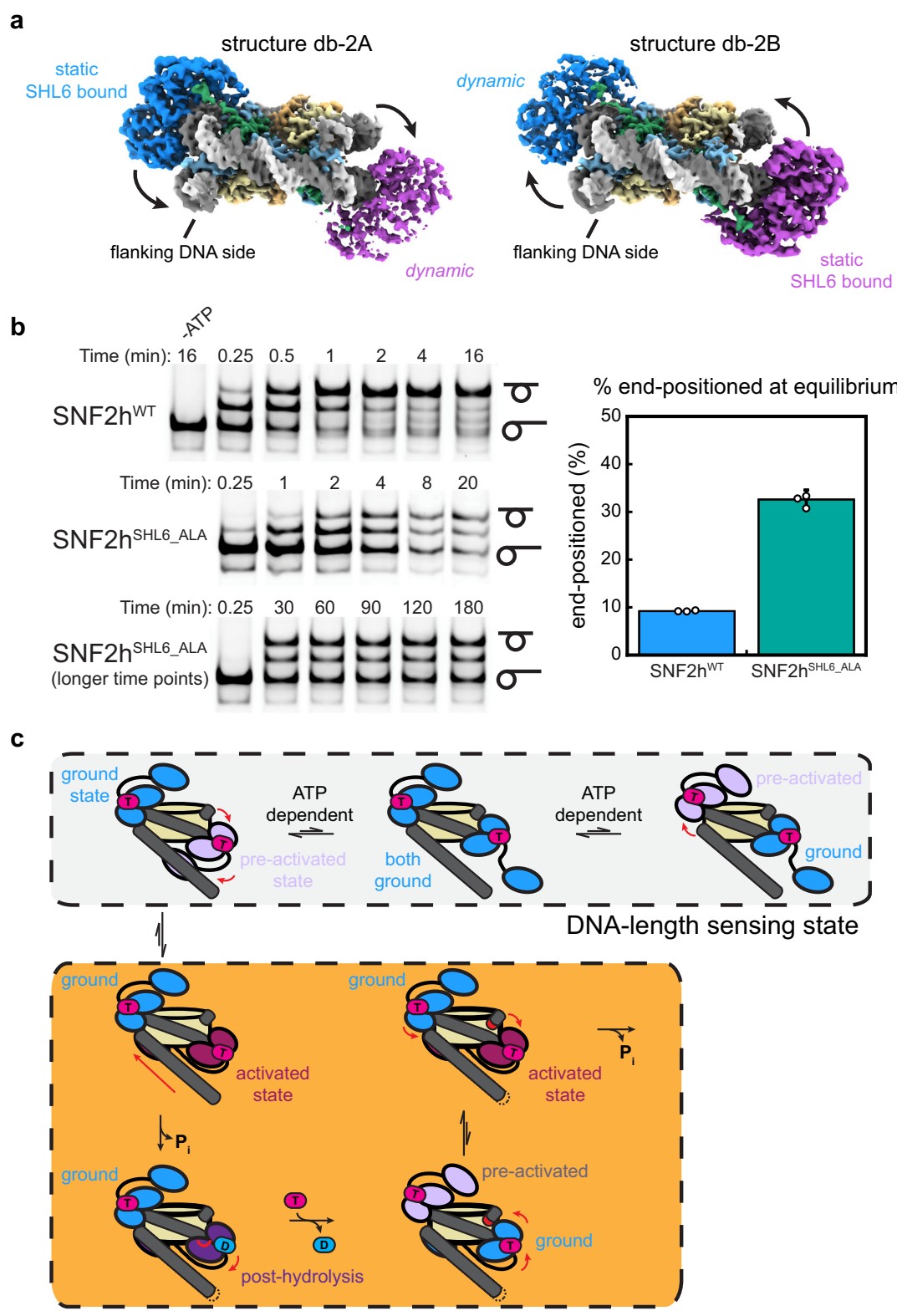

To determine the importance of SHL6 contacts on SNF2h coordination, we tested the nucleosome remodeling activity of the SNF2h$^{SHL6\_ALA}$ mutant using a native gel mobility assay. Wild-type SNF2h remodels end-positioned nucleosomes with 60 base pairs of flanking DNA such that only ~10% of nucleosomes remain end-positioned at steady state (Fig. 4b). In contrast, we observe a different distribution of nucleosome products for SNF2h$^{SHL6\_ALA}$, such that ~33% of nucleosomes remain end-positioned (Fig. 4b). While the slower rate for nucleosome remodeling is expected given the defect in ATP hydrolysis rate with SNF2h$^{SHL6\_ALA}$ (Fig. 3c), the defect in nucleosome centering even after completion of the reaction suggests that SHL6 contacts also play an important role

**Fig. 4 | Coordinated asymmetric actions of SNF2h protomers on a nucleosome.** **a** Two coulomb potential maps representing the endpoints of one principal component from 3D variability analysis of the double-bound SNF2h-nucleosome particles. On one end (structure db-2A), one SNF2h is stably bound to the nucleosome, while the other SNF2h is more dynamic. On the other end (structure db-2B), the SNF2h that was stably bound to the nucleosome becomes more dynamic, while the SNF2h that was more dynamic becomes more static. **b** (left) Nucleosome remodeling assays visualizing SNF2h-mediated centering of end-positioned nucleosomes using native PAGE. Top and bottom symbols to the right of the gels denote centered and end-positioned nucleosomes, respectively. In the experiments shown, [nucleosome] = 15 nM, [SNF2h] = 500 nM, and [ATP] = 4 mM. The bottom two time courses are both with the mutant SNF2h[SHL6_ALA], with one containing longer time points. Source data are provided as a Source Data file. (right) Quantification of

the percent of nucleosomes that remain end-positioned at equilibrium for wild-type SNF2h and SNF2h[SHL6_ALA]. Data are presented as mean values +/- SD. $n = 3$ independent nucleosome remodeling experiments. Source data are provided as a Source Data file. **c** Model for SNF2h conformations while coordinating actions on a nucleosome based on all available data. In the DNA-length sensing state, each SNF2h protomer can either be in a static ground state or dynamic pre-activated state while sensing for flanking DNA using its HAND-SANT-SLIDE (HSS) domain. The protomer that is able to sense flanking DNA will undergo further conformational change to reach an activated state that promotes ATP hydrolysis and DNA translocation. Exchange of ADP for ATP resets SNF2h to a ground state, and the other SNF2h protomer then has first priority to again search for flanking DNA in a dynamic, pre-activated state.

in coordinating remodeling activities between SNF2h protomers to properly center nucleosomes.

Previous single-molecule FRET studies showed that SNF2h acts on nucleosomes in a manner where the enzyme senses flanking DNA length during translocation pauses followed by rapid ATP-dependent DNA translocation steps[24,41–44]. We propose that the structures observed here represent static snapshots of the SNF2h protomers taking turns in the act of sensing flanking DNA length (Fig. 4c), although we do not directly observe density in our structures for the DNA length-sensing HSS domains on SNF2h. Our findings are consistent with previous models, in which stable association of the HSS domain with flanking DNA drives the particular SNF2h protomer to hydrolyse ATP, concomitant with repositioning of the HSS domain (Fig. 4c)[23].

## Discussion

Cryo-EM sample preparation for chromatin complexes remains a notoriously challenging problem. Complexes tend to dissociate upon plunge freezing due to denaturation at the AWI, which prevents reproducible grid preparation. To address this issue, we have developed ssDNA- and TAASTY-functionalized GO grids that are able to move a SNF2h-nucleosome sample away from the AWI through non-specific charge interactions and permit reproducible preparation of suitable cryo-EM grids. We have previously reported a facile way of layering GO onto EM grids without the need for specialized equipment[25], and the functionalization here only adds an incubation and wash step to the previous procedure. Similar to other grids with a support film, we find that signficantly less sample is needed for the functionalized GO grids than for conventional Quantifoil grids (-10-fold less). The grids allow for the collection of large amounts of useful data, and due to the protective property of the grids, we find that a larger percentage of SNF2h-nucleosome particles that were picked could be retained for downstream processing versus the need to discard bad denatured particles using conventional Quantifoil grids.

With data collected from the functionalized GO grids, we were able to determine high-resolution structures of the SNF2h-nucleosome complex without the need to fix the sample with crosslinking. The resolution of the consensus map at -2.3 Å is comparable to the highest resolution nucleosome-remodeler structure determined to date with crosslinking[7]. We were able to apply 3D variability analysis[31] to determine high-resolution maps corresponding to various nucleosome and SNF2h motions. In addition to intrinsic nucleosome motions such as nucleosome squeezing/expansion and DNA unwrapping similar to motions that have been previously observed[29,30,36], we also see SNF2h rocking on the nucleosome at high resolution with the single-bound SNF2h-nucleosome particles. In one state, SNF2h stably engages nucleosomal DNA at SHL6 bound with clear density for ADP-BeFx, similar to previously determined structures of SNF2h or the yeast homolog ISW1 bound to nucleosome in the presence of ADP-BeFx[14,39]. In the other state, both lobes of the SNF2h ATPase domain rock upward, and SNF2h is detached from nucleosomal DNA at SHL6 and

we detect ADP in the active site without density for BeF$_X$. Considering: 1. the ATP hydrolysis defect exhibited by mutant SNF2h[SHL6_ALA] and previous mutagenesis data that indicate Snf2 interaction with SHL6 is important for ATPase activity[40]; and 2. a previous ADP-bound ISW1-nucleosome structure that showed an upward shift of the second ATPase lobe in conjunction with partial DNA translocation[39], we hypothesize that SNF2h dissociation from SHL6 represents an intermediate state that is closer to the post-hydrolysis ADP•P$_i$ state and that complete SNF2h dissociation from SHL6 shifts SNF2h to an ADP state concomitant with P$_i$ release (Fig. 3d; step 1). In this model, during completion of the ATP hydrolysis cycle, the top lobe of SNF2h further shifts upwards and DNA is partially translocated (Fig. 3d; step 2). We speculate that engagement of SHL6 promotes exchange of ADP for ATP, which in turn allows for the complete translocation of DNA and resets SNF2h for subsequent ATPase cycles (Fig. 3d; step 3), in agreement with a previously proposed model where ATP binding completes DNA translocation[45]. We note that the SNF2h[SHL6_ALA] mutant mainly tests the importance of contacting SHL6 and further experiments are needed to test the specific hypothesis that SNF2h dissociation from SHL6 represents an intermediate state.

Applying 3DVA to the double-bound SNF2h-nucleosome particles, we were able to see a coordinated, asymmetric motion between the two SNF2h protomers on the nucleosome, where one protomer dissociates from SHL6 and becomes more dynamic while the other protomer is SHL6-bound and static. Mutant SNF2h[SHL6_ALA] appears to be defective in properly centering nucleosomes, suggesting SHL6 contacts are important for coordinating remodeling activities between SNF2h protomers. This asymmetric coordination resonates with previously proposed models where the two SNF2h protomers are able to take turns acting on a nucleosome[23,24]. Each protomer is able to use their HAND-SANT-SLIDE (HSS) domains to sense for flanking DNA during translocation pauses as observed by single-molecule FRET[24,41–44], and stable engagement of one protomer's HSS domain for flanking DNA promotes SNF2h to move towards a conformation primed for ATP hydrolysis[23]. Thus, the functionalized GO grids enabled us to see conformational states of SNF2h that we hypothesize represent states during the DNA-length sensing stage. We note that while we extensively classified particles to align for flanking DNA, due to the higher flexibility and thus lower resolution of the DNA used for alignment, it remains a possibility that particles are still misaligned due to the pseudo-C2 symmetry of the sample. Therefore, an alternate possibility is that SNF2h is more dynamic only on one particular side of the nucleosome, possibly influenced by asymmetry in the Widom 601 nucleosome positioning sequence as was shown to affect remodeling by the Chd1 chromatin remodeler[46].

In addition, double-bound structures of other nucleosome-remodeler complexes have also been observed, including Snf2[40] and ALC1[47]. However, unlike for SNF2h, which has been shown to remodel nucleosomes cooperatively with a Hill coefficient of approximately two[22] and remodel nucleosomes at a faster rate when the two protomers are covalently connected[23], there is no evidence to suggest that

the two protomers for Snf2 and ALC1 can coordinate their remodeling. There may not be a need for Snf2 protomers to coordinate their remodeling since Snf2 is part of the SWI/SNF complex[8] in vivo, which consists of many other subunits that likely preclude two SWI/SNF complexes from engaging the same nucleosome substrate. Similarly, ALC1 prefers to engage poly ADP-ribosylated (PARylated) nucleosomes[48], and these nucleosomes may only be modified on one face. Therefore, nucleosome PARylation may instead determine whether ALC1 protomers can act as a dimer.

Despite the advantages provided by the functionalized GO grids, there remain issues that the grids do not necessarily address. For a nucleosome-only sample, we observed strong preferred orientation that precluded determination of a 3D map, which is not usually a problem with traditional Quantifoil holey-carbon grids[28–30]. Therefore, the grids may cause preferred orientation for certain samples, likely caused by specific interaction between the sample and the functionalized GO substrate, although there was no preferred orientation issue for the SNF2h-nucleosome complex. In addition, previous NMR studies indicate that histone distortions play an important role in permitting DNA translocation around nucleosomes by SNF2h, and chemical shift perturbations for residues in histones H3 and H4 were observed upon SNF2h binding in the presence of ADP-BeFx[49]. However, when comparing structures determined from 3DVA maps, no significant RMSD differences between the same residues were observed. Therefore, there are still limitations in our ability to use single-particle cryo-EM to see conformational changes that are observable by NMR.

We anticipate that the GO grids can also be functionalized with other DNA sequences and/or polymers with other charge properties for different samples. We hypothesize that the charged, functionalized grids will facilitate cryo-EM sample preparation of other protein-nucleic acid complexes and possibly other proteins and complexes with charged surfaces, as well.

## Methods

### Nucleosome reconstitution
Histones from *Xenopus laevis* were recombinantly expressed and purified from *E. coli*[50]. Histone octamers were then reconstituted using purified histones[50,51]. The Widom 601 nucleosome positioning sequence[52] with an extra 60 base pairs of arbitrary DNA flanking the 3'-end was used for nucleosome assembly using salt gradient dialysis and purified through a 10- 30% glycerol gradient[51]. For Cy5-labeled nucleosomes for native gel remodeling assays, the DNA was generated through PCR using a Cy5-labeled forward primer (/5Cy5/ CTGGAGAATCCCGGTGCCG from IDT) on the DNA exit side in the context of nucleosome remodeling. The reverse primer sequence used for PCR is AGAGTGGGAGCTCGGAACAC.

The sequence used for nucleosomal DNA in this study is as follows:

CTGGAGAATCCCGGTGCCGAGGCCGCTCAATTGGTCGTAGACAG CTCTAGCACCGCTTAAACGCACGTACGCGCTGTCCCCCGCGTTTTAA CCGCCAAGGGGATTACTCCCTAGTCTCCAGGCACGTGTCAGATATAT ACATCCTGTGCATGTATTGAACAGCGACCTTGCCGGTGCCAGTCGG ATAGTGTTCCGAGCTCCCACTCT

### SNF2h expression and purification
SNF2h was recombinantly expressed and purified from *E. coli*[23]. Briefly, N-terminally 6xHis-tagged SNF2h was affinity purified from lysate using TALON resin. TEV protease was then used to remove the 6xHis-tag. The protein was passed through a HiTrap Q column to remove contaminating DNA and further purified using a Superdex 200 size exclusion column.

SNF2h$^{SHL6\_ALA}$ was generated by site-directed mutagenesis, mutating $^{292}$KEKSVFKK$^{299}$ to $^{292}$AEAAVFAA$^{299}$. SNF2h$^{SHL6\_ALA}$ expressed and purified similarly to wild-type SNF2h with no notable differences.

### ATPase assay
ATPase reactions were performed under single turnover conditions with respect to ATP (i.e. enzyme in excess of ATP, but nucleosomes are in excess of enzyme). Reactions were performed at 20 °C with 15 nM SNF2h, 80 or 320 nM nucleosome, and trace amounts of γ-$^{32}$P-ATP in assay buffer (12.5 mM HEPES-KOH pH 7.5, 70 mM KCl, 3 mM MgCl$_2$, 0.02% NP-40). Reactions were started with addition of enzyme, and 2 μL time points were quenched with an equal volume of quench buffer (50 mM Tris-HCl pH 7.5, 3% SDS, 100 mM EDTA pH 8). Inorganic phosphate was resolved from ATP on a Baker-flex PEI-cellulose TLC plate (JT Baker/Avantor) with 0.5 M LiCl/1 M formic acid mobile phase. Plates were dried, exposed to a phosphorscreen overnight, and scanned using a Typhoon imager (Cytiva). Rate constants were determined using KaleidaGraph v4.0 by linearly fitting the initial reaction time courses (i.e. <10% complete).

### Native gel remodeling assay
Remodeling reactions were performed under single turnover conditions with respect to nucleosome (i.e. enzyme in excess of nucleosome, but ATP is in excess of enzyme). Reactions were performed at 20 °C with 15 nM Cy5-labeled nucleosome, 500 nM SNF2h, and 4 mM ATP•MgCl$_2$ in assay buffer. Reactions were started with addition of enzyme, and 5 μL time points were quenched with an equal volume of stop solution (40 mM ADP, 0.8 mg/mL pUC19 plasmid, 8% glycerol) that contains excess ADP and plasmid DNA. Time points were then resolved by native PAGE (6% acrylamide, 0.5X TBE) and scanned using a Typhoon imager (Cytiva). Gels were then quantified using Image-Quant TL v8.2. The percent of nucleosomes end-positioned at equilibrium was determined by the percent of fast-migrating nucleosomes to the total nucleosome intensity.

### TAASTY synthesis
TAASTY was made from a 1:1 copolymer of styrene and acrylic acid, poly(acrylic acid-co-styrene), made through reversible addition fragmentation chain transfer (RAFT) polymerization, with subsequent hydrolysis of the terminal trithiocarbonate. Poly(acrylic acid-co-styrene) was prepared as previously described[53]. In short, an equimolar amount of acrylic acid and styrene was copolymerized using 2-(Dodecylthiocarbonothioylthio)−2-methylpropionic acid as the RAFT agent and azobisisobutyrunitrile as the initiator. The ratio of monomer/RAFT/initiator was 100/1/0.2. Both monomers were distilled prior to use, and the reaction mixture was sparged with nitrogen prior to heating. The neat reactant mixture was heated to 60 °C for 6 h, at which point the reaction had reached a total monomer conversion of 95% by H1-NMR. The yellow solid was dissolved in diethyl ether, and precipitated into hexane. The yellow powder was recovered by filtration, and dried in vacuo, yielding poly(acrylic acid-co-styrene), Mn = 6.8 kDa, Mw = 8.4 kDa, as measured by SEC-MALS. The removal of the RAFT trithiocarbonate was achieved by dissolving 1 g of the copolymer in 5 mL of a 1:1 mixture of 1 M NaOH and ethanol, subsequently adding 20 eq of butylamine and TCEP. This reaction mixture was stirred for 16 h at room temperate. This yielded a colorless, clear solution, which was diluted into 40 mL of ultrapure water in a centrifuge tube. This mixture was acidified with 5 M HCl(aq), until pH was below 1, forming a white precipitate. The precipitate was spun down, and the supernatant discarded. The precipitate was then dried in vacuo, and washed and triturated with a 1:1 ether:hexane mixture, and dried in vacuo.

### GO grid preparation
Freshly oxidized GO was layered onto gold Quantifoil R1.2/1.3 300 mesh grids[25]. Briefly, grids were plasma cleaned for 10 s with argon gas. Grids were placed onto a mesh submerged in ultrapure water in a petri dish. GO diluted to 0.4 mg/mL in dispersant solution (5:1 methanol: water) was applied dropwise surrounding the grids. Water was slowly

pumped out from the bottom of the dish. Grids were allowed to dry overnight at room temperature in the dark.

## GO grid functionalization

ssDNA oligo with a 5' amino modifier C6 was ordered from Integrated DNA Technologies (5/AmMC6/GGTACCCGGGGATCG) and diluted to 0.2 mM in DMSO. Each freshly prepared GO grid was submerged in 30 μL ssDNA solution in a 1.5 mL microcentrifuge tube shaking at 300 rpm at 24 °C in a covered ThermoMixer overnight for ssDNA functionalization.

TAASTY co-polymer was dissolved at a final concentration of 10 mg/mL in DMF with triethylamine. Each freshly prepared GO grid was layered on top of a 30 μL droplet of TAASTY solution at room temperature for 2–4 h for TAASTY functionalization.

Each functionalized GO grid was rinsed with water and then 100% ethanol and allowed to dry on filter paper. Functionalized GO grids were stored at −20 °C in the dark until use. Grids were typically used within 1–2 days, and the exact shelf life of the grids has not been determined.

## Cryo-EM sample preparation and data collection

Nucleosome-only cryo-EM grids were prepared with 50-100 nM 601-positioned nucleosomes with 60 base pairs of flanking DNA on one end (0/60 nucleosomes) in EM buffer (12.5 mM HEPES-KOH pH 7.5, 60 mM KCl, 3 mM $MgCl_2$, 1.5% glycerol). SNF2h-nucleosome cryo-EM grids were prepared by first mixing 100 nM 0/60 nucleosomes with 500 nM SNF2h in EM buffer supplemented with 2 mM ADP-BeF$_x$ (2 mM ADP, 2 mM $MgCl_2$, 2 mM $BeSO_4$, 10 mM NaF). The sample was incubated at room temperature for 30 min prior to plunge freezing.

Sample grids were plunge frozen using a Vitrobot set at 4 °C and 100% humidity following a protocol similar to one previously described[54]. Functionalized GO grids were first held on Vitrobot tweezers outside the Vitrobot. 2.5 μL of sample was applied to the grid and allowed to incubate for 45 s. The grid was side-blotted with Whatman 1 filter paper and then rinsed twice with droplets of buffer on parafilm before side-blotted again. A fresh 2.5 μL of buffer was applied to the grid and then the tweezers were placed onto the Vitrobot. The grid was blotted with humidity-saturated Whatman 1 filter papers for 3 s with a blot force of −1 before being plunge frozen into liquid ethane. Nucleosome-only datasets were collected on a 200 kV FEI Talos Arctica at UCSF. Single-particle SNF2h-nucleosome datasets were collected on a 300 kV Titan Krios at UCSF using a K3 camera at a magnification of 105kx (0.84 Å/pix). Electron cryotomography data on the SNF2h-nucleosome samples were collected on a 300 kV Titan Krios at UCSF using a K3 camera at a magnification of 64kx (1.37 Å/pix). Data was acquired with a dose-symmetric tilt-scheme collecting data at every 3 degrees in a tilt range ±60°. All data was collected using Serial EM v3.7 or newer.

## Cryo-EM data processing for SNF2h-nucleosome datasets

For the SNF2h-nucleosome datasets, raw movies were imported into RELION v3.1[55] and motion-corrected using UCSF MotionCor2 v1.4.1[56] within RELION. Dose-weighted micrographs were imported into cryoSPARC v3.3.2[57] and patch CTF estimation (multi) was used to estimate defocus values. A nucleosome map was used to generate templates for template-picking of particles, and a total of ~8.5 million particles were extracted. UCSF Chimera v1.16.0 was used to visualize maps during data processing. Ab-initio reconstruction was performed with extracted particles and terminated early to generate 3 'junk' classes. Heterogeneous refinement was performed with a good nucleosome map and the 3 junk maps as templates. This process was repeated for a total of 3 rounds of heterogeneous refinement. After classification, ~2 million particles remained in the good SNF2h-nucleosome class. These particles were taken back to RELION[58] using UCSF pyem v0.5 for Bayesian polishing and CTF refinement, and then

back to cryoSPARC for non-uniform refinement which resulted in a 2.3 Å consensus reconstruction.

To separate single-bound particles from double-bound particles, skip-align 3D classification was performed in RELION with a spherical mask for SNF2h. Two out of four classes had density for SNF2h, where one class had clear SNF2h density and the other class had weak SNF2h density. A second round of skip-align 3D classification was performed for the first class of particles (655,539 particles) with a spherical mask for possible SNF2h on the opposite SHL2 position. One class had clear density for SNF2h and represented double-bound particles (39,882 particles). One class had the majority of particles (544,890), which suggested the possibility that not enough classes were available for proper classification. Therefore, another round of skip-align 3D classification was performed for this set of particles, which resulted in a class of single-bound particles (324,612 particles) and a class of double-bound particles (112,806 particles).

From the first round of skip-align 3D classification, the class with weak SNF2h density was taken for another round of skip-align 3D classification with a mask for SNF2h at the same position. While all four classes showed density for SNF2h, the two most populated classes were taken for skip-align classification with a spherical mask for possible SNF2h on the opposite SHL2 position. These classifications resulted in classes with double-bound particles (104,929 particles and 24,518 particles) and single-bound particles (304,790 particles, 47,713 particles, and 32,357 particles). Therefore, a total of 709,472 single-bound particles and 282,135 double-bound particles were isolated from these rounds of classification. Additional rounds of skip-align classification with spherical masks for SHL2 positions were performed on the discarded particles (992,245 particles) to recover more particles. These rounds of classification resulted in 252,585 more single-bound particles and 58,941 more double-bound particles, for a total of 962,057 single-bound particles, 341,076 double-bound particles, and 680,719 unbound, borderline, and/or junk particles.

To find the position of the 60-bp flanking DNA of the nucleosome for the single-bound particles, the 962,057 single-bound particles were first refined applying C2 symmetry and then symmetry expanded using RELION. Skip-align focused classification in RELION was performed with a spherical mask for the position of flanking DNA. One class with 36.9% of the particles had clear density for flanking DNA. Subsequent skip-align focused classifications in RELION were performed with spherical masks for SNF2h at the SHL+2 and SHL-2 positions, which resulted in 178,672 particles with SNF2h at SHL+2 and 148,741 particles with SNF2h at SHL-2 after duplicate removal. However, while there was clear flanking DNA density for the SNF2h at SHL-2 map at lower contour, the flanking DNA density was more ambiguous for the map with SNF2h at SHL+2. Therefore, we directly looked at the density for DNA bases within the map, which were sufficiently high in resolution to distinguish between purines and pyrimidines in most cases. Specifically, for a region of DNA where purines and pyrimidines would be swapped depending on the orientation of the DNA, the DNA model matched the density well for the SNF2h at SHL-2 map. However, in the same region, the DNA model did not match the density well for the SNF2h at SHL+2 map, and instead matched better if the DNA model was modeled with the opposite orientation. Therefore, the SNF2h at SHL+2 map also represents SNF2h at SHL-2 instead. Refinement of all single-bound particles together results in a map with flanking DNA visible at low contours with SNF2h at SHL-2. Inspection of the DNA base pair densities also agree with the conclusion that SNF2h is at the SHL-2 position.

To find the position of the 60-bp flanking DNA of the nucleosome for the double-bound particles, the 341,076 double-bound particles were used to generate maps ab initio with C2 symmetry applied in cryoSPARC and then the one good class was symmetry expanded using RELION. Skip-align 3D classification was performed in RELION with a spherical mask for flanking DNA. The largest class with 45.9% of the

particles had no flanking DNA density and was taken for local refinement in cryoSPARC after duplicate removal, which revealed flanking DNA density on the other side. After a round of 3D variability analysis, we observed that in the first principal component, one endpoint map had density for flanking DNA on both sides, which suggested certain particles were still either misaligned and/or ambiguous for flanking DNA position. We therefore used the intermediates function in cryoSPARC to remove the ambiguous particles and repeated 3D variability analysis. Ambiguous particles were still present, and we repeated the procedure for a total of 3 rounds of 3DVA. After the fourth round of 3DVA, we no longer saw any endpoint map with flanking DNA on both sides with 107,768 particles remaining.

### Cryo-EM data processing for nucleosome-only dataset

For the nucleosome-only dataset, raw movies were motion-corrected using UCSF MotionCor2 v1.4.1 and the resulting dose-weighted micrographs were imported into cryoSPARC. Patch CTF estimation (multi) was used to estimate defocus values. A nucleosome map was used to generate templates for template picking of particles. Ab-initio reconstruction was performed with extracted particles and terminated early to generate 3 'junk' classes. Heterogeneous refinement was performed with a good nucleosome map and the 3 junk maps as templates. This process was repeated for a total of 3 rounds of heterogeneous refinement. 2D classification was performed with the particles in the final good class for the 2D classes shown in Fig. 1. Notably, while we were able to see good nucleosome 2D classes, we were not able to generate a high-resolution reconstruction from the nucleosome-only dataset due to preferred orientation.

### SNF2h-nucleosome tomographic analysis

For SNF2h-nucleosome tilt series, raw movies were motion-corrected using UCSF MotionCor2 v1.4.1 and the resulting micrographs were imported into AreTomo v1.4.2 for marker-free tilt series alignment and tomogram reconstruction[59]. Contrast in the tomograms was improved using IsoNet v0.2.1[60].

### Model building

cryoSPARC maps were used to build models for the single- and double-bound SNF2h-nucleosome structures. The previously determined model for the SNF2h-nucleosome complex (PDB 6NE3) was used as the initial template. The DNA was corrected for the lack of 2 base pair translocation in the current maps. Iterative corrections in COOT v0.9.6 and real space refinement in Phenix v1.18.2 were then used to fix differences in positions of amino acids between the current structures and the previous structure. The quality of all refined models was assessed using model validation in Phenix and the wwPDB validation server.

### Reporting summary

Further information on research design is available in the Nature Portfolio Reporting Summary linked to this article.

## Data availability

The atomic coordinates generated in this study have been deposited to the RCSB Protein Data Bank under accession codes 8V4Y (SNF2h-nucleosome single-bound structure 1), 8V7L (SNF2h-nucleosome single-bound structure 2), and 8V6V (SNF2h-nucleosome double-bound structure). The cryo-EM Coulomb potential maps generated in this study have been deposited in the Electron Microscopy Data Bank under accession codes EMD-43000 (SNF2h-nucleosome highest resolution consensus map), EMD-43001 (SNF2h-nucleosome consensus single-bound map), EMD-42977 (SNF2h-nucleosome single-bound map 2 A), EMD-43003 (SNF2h-nucleosome single-bound map 2B), EMD-43002 (SNF2h-nucleosome consensus double-bound map), EMD-43004 (SNF2h-nucleosome double-bound map 2 A), and EMD-43005 (SNF2h-nucleosome double-bound map 2B). The raw movies generated in this study for the SNF2h-nucleosome complex have been deposited to EMPIAR under accession code EMPIAR-11909. The ATPase assay and nucleosome remodeling assay data generated in this study are provided in the Source Data file. Source data are provided with this paper.

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

## Acknowledgements

We thank Nathan Gamarra and Laura Hsieh for nucleosome samples for initial experiments, Maxine Bi and Shawn Zheng for assistance with cryo-ET, Julia Tretyakova and Yongqiang (John) Wang for managing the Narlikar and Cheng labs respectively, Junrui Li, Chengmin Li, and Matt Harrington for maintaining computational resources, David Bulkley and Glenn Gilbert for maintaining the UCSF EM facility, and Gregory D. Bowman, Ilana Nodelman, and members of both the Cheng and Narlikar labs for critical discussions. This work was supported by grants from the National Institute of Health (R35GM140847 to Y.C. and R35GM127020 to G.J.N.) and a NIH NIGMS fellowship F32GM137463 to U.S.C. A.A.A.A. was supported by grants NNF18OC0030896 from the Novo Nordisk Foundation and the Stanford Bio-X program and 0171-00081B from Independent Research Fund Denmark. H.E.A was funded by grant R265-2017-4015 from the Lundbeck Foundation. The UCSF cryo-EM facility was partially supported by NIH grants (S10OD020054, S10OD021741, and S10OD025881). Y.C. is an investigator of the Howard Hughes Medical Institute. The figures for this manuscript were generated using UCSF ChimeraX v1.4. UCSF ChimeraX is developed by the Resource for Biocomputing, Visualization, and Informatics at the University of California,

San Francisco, with support from National Institutes of Health R01-GM129325 and the Office of Cyber Infrastructure and Computational Biology, National Institute of Allergy and Infectious Diseases.

## Author contributions

E.P. initiated the project for functionalizing GO grids with ssDNA. E.P. and E.N.M. performed initial experiments with nucleosomes with complementary DNA. U.S.C. and E.P. prepared cryo-EM samples for the SNF2h-nucleosome complex using functionalized GO grids. U.S.C. and E.P. collected cryo-EM data with assistance from Z.Y. E.P. performed preliminary cryo-EM data analysis. U.S.C. performed subsequent cryo-EM data analysis with guidance from J.P.A. and Y.C. U.S.C. collected and analyzed tomography data. E.P. and H.A. conceived the idea for polymer-functionalized GO grids. A.A.A.S. synthesized TAASTY copolymer. F.W. synthesized GO, and F.W. and D.A. provided support on use of GO and on functionalization. U.S.C., G.J.N., and Y.C. interpreted final structures. U.S.C. designed, performed, and analyzed all biochemical experiments with input from G.J.N. U.S.C. drafted the manuscript and figures with input and revisions from E.P., H.A., A.A.A.S., G.J.N., and Y.C. All authors approved the final version of the manuscript.

## Competing interests

Y.C. is on the scientific advisory board of ShuiMu BioSciences. The remaining authors declare no competing interests.
