## [Peer Review File · Nature Communications]

Functionalized graphene-oxide grids enable high-resolution cryo-EM structures of the SNF2h-nucleosome complex without crosslinkingReviewers' Comments:

Reviewer #1:

Remarks to the Author:

Air-water interface (AWI) has been demonstrated as a major obstacle to overcome to enhance the efficiency of cryo-EM structural determination of macromolecules. In the current work by Chio et al., the authors present a novel approach to functionalize graphene-oxide covered EM grids with specific ssDNA or TAASTY co-polymer to anchor chromatin remodeler SNF2h and nucleosome therefore preventing them from adsorbed to the AWI during cryo-EM specimen preparation. The authors reported successful preservation of complexes for high-resolution 3D cryo-EM structural determination, suggesting the effectiveness of the method's applicability for high-resolution cryo-EM of macromolecular complexes. Using the above methods, the authors successfully solved the structure of SNF2h and its complex with nucleosomes in various conformational and assembly states, providing important insights to the SNF2h's activity on nucleosomes. This work is written clearly with solid evidence to support the design and the discovery. This work will be of great interest to the field of cryo-EM and structural biology and biochemistry on protein-DNA complexes.

I recommend the work may be published with the following suggestions.

1. The authors should provide cryo-electron tomography analysis of the specimen to demonstrate the particle distribution in the vitreous ice. This will help understand whether the novel method has true effect to protect complexes from AWI.
2. The authors mentioned the strong preferential orientation of nucleosome only sample on the functionalized GO grid. Was this on ssDNA- or TAASTY- functionalized GO? The authors may elaborate more on the potential various chemical property's effect on particle orientation.
3. The authors described a few times of "near perfect" cryo-EM grids. Only if the authors have a statistical data to show the significantly better performance on the reproducibility and successful rate of the specimen, it is dangerous to claim a "perfect" grid.
4. Fig. 3b, the two magnesium ions in the nucleotide density may be more clear with a higher thresholded density map.

Reviewer #2:

Remarks to the Author:

The manuscript titled "Functionalized graphene-oxide grids enable high-resolution cryo-EM structures of the SNF2h-nucleosome complex without crosslinking" by Un Seng Chio et al. reports two new functionalized graphene-oxide support films for cryo-EM which aid in overcoming the air-water denaturation issue often encountered during cryo-EM sample preparation. The authors utilize these functionalized graphene-oxide grids to determine structures of chromatin remodeler SNF2h bound to nucleosomes in the absence of crosslinking.

In previous studies, SNF2h is suggested to function optimally as a dimer and negative stain EM structures have been able to capture two SNF2h monomers bound to nucleosomes. However, high-resolution structural characterization of the dimeric form SNF2h interacting with nucleosomes has not been successful due to limitations in sample preparation (crosslinking) and air-water denaturation issues. The authors have developed new functionalized graphene oxide grids using either ssDNA or a similar negatively charged polymer (TAASTY) to address the air-water denaturation issue. Utilizing these functionalized grids, the authors determined high-resolution structures of both single SNF2h-bound and double SNF2h-bound nucleosomes. Using particle sorting and 3D variability analysis, the authors could discern multiple states of SNF2h bound to nucleosomes for both the single-bound and double-bound configurations allowing. The authors propose a model based on these structures which explains why SNF2h functions optimally as a dimer. In summary, the authors have come up with an interesting strategy for visualizing a challenging nucleosome-bound complex. Overall, the manuscript is technically sound with an interesting new GO-based grid preparation strategy to address the air-

water interface, and the authors have managed to visualize the double-bound SNF2h to nucleosome at a higher resolution compared to their previous study. However, the authors have not provided any functional assays to support the ATP hydrolysis model proposed as well as the coordinated action of the two SNF2h protomers. Therefore, it is difficult for this reviewer to relate the conformational changes visualized to the SNF2h translocation of DNA.

The main structural observations are:

1. The authors were able to visualize ADP-BeFx in the single-bound SNF2h to nucleosomes whereas in the previous structure of SNF2h bound to nucleosomes, only ADP was visualized. The authors were also able to visualize conformational changes in SNF2h with one state bound to ADP-BeFx and another with ADP bound (see related point # 3 below in comments).
2. No major conformational changes within the nucleosome in both single-bound and double-bound SNF2h. Although the authors were able to tease apart DNA unwrapping with or without the loss of density for H3, and H2A using 3D variability analysis in some states.
3. Although the authors have previously visualized double-bound SNF2h to nucleosomes at lower salt concentrations (70 mM), these reconstructions were at lower resolution $\sim 8 \text{ \AA}$. Here, the authors have managed to visualize the double-bound SNF2h to nucleosome with high-resolution structures of the SNF2h protomers. However, the authors were only able to observe ADP bound to both these protomers in the double-bound SNF2h to nucleosomes. 3D variability analysis of this double-bound SNF2h to nucleosomes suggests an interesting asymmetric conformational flexibility. However, the resolution of the protomers from the variability components precluded the unambiguous identification of nucleotide states.
4. The authors have hence inferred nucleotide states based on the engagement of the protomer with SHL+6 of the nucleosome and correlated it to the conformations observed in the single-bound SNF2h to nucleosomes. In the double-bound SNF2h structure and the corresponding variability components, the authors infer that SNF2h stably bound to SHL+6 must have ADP-BeFx whereas SNF2h not engaged with SHL+6 has ADP.

Comments:

1. The authors do not directly demonstrate that the ssDNA and TAASTY functionalized GO grids retain particles away from the air-water interface. Could the authors demonstrate this using a tilt-series collection?
2. The authors also do not comment on the broad use of such functionalization and whether the use of these functionalizations is restricted to nucleosome or DNA-bound protein complexes. Although this reviewer feels it might not be unnecessary to demonstrate their use on a protein that isn't bound to DNA or RNA, it would be helpful for the readers to understand their broad use if the authors comment on this.
3. Since the nucleosome alone sample exhibited strong preferential orientation as stated by the authors, I request the authors to provide 3D FSC for all SNF2h-nucleosome maps for the readers to better assess the structures generated. In addition, could the authors comment on whether the nucleosome maps with TAASTY GO-grids also showed similar preferential orientation?
4. On page 5, line 128, the authors state that it is impossible to observe intact nucleosomes, even at concentrations of 1 μ M, without cross-linking on regular Quantifoil grids. However, this statement should perhaps be reworded as imaging of nucleosomes without crosslinking (with phase plate) has yielded high-resolution (<https://doi.org/10.1093/nar/gkw708>). However, the reviewer appreciates the intent of the original statement but requests that it be reworded to reflect the author's experience.

5. It is important for the authors to provide clarification regarding the physiological salt concentration throughout the manuscript. In line 125, they assume that 150mM KCl represents a relevant concentration required for chromatin remodeling enzymes bind to the nucleosome. However, the data provided here is at 60 mM KCl and the previous studies were carried out at 140 mM KCl and 70 mM KCl. Could the conformational variability of the flanking DNA as well as the conformational flexibility observed in this study be due to the lower salt concentration overall? The authors in Extended Fig. 6 show that DNA conformational flexibility is clearly different at 140 mM KCl versus 60 mM KCl, how much influence does this flexibility have on SNF2h binding and the associated conformational changes visualized? The authors should perhaps visualize these interactions at 140 mM KCl to gauge functional relevance but since this hasn't been presented it remains difficult to correlate the observations with a functional model.

6. Could the authors show a better figure showing ADP-BeFx in the variability analysis component for SNF2h bound to SHL+6? The current Fig. 3b is showing with Mg²⁺ and BeFx but it isn't clear based on the reference to Fig. 3b in text (Line 229). Perhaps a figure like that shown in Extended Figure 5b showing ADP and ADP-BeFx for this variability analysis component would be more illustrative for the readers.

7. One of the major conclusions from the authors that the asymmetric motion of the two SNF2h protomers are coordinated and could be related to DNA flanking length sensing and nucleotide hydrolysis is speculative as the authors do not provide any direct evidence for this conclusion. Since the authors also do not observe the DNA-length sensing HSS domains, it remains unclear how SNF2h conformation together with nucleotide state is linked to the translocation cycle.

8. The authors propose a hypothesis that SNF2h dissociation from SHL+6 is an intermediate state in the ATP hydrolysis cycle based on the conformations observed here as well as a comparison with other remodeler enzymes such as ISWI. Since the authors do not directly provide evidence for this hypothesis the conclusion hence remains a speculation in the manuscript as the authors have themselves stated. This reviewer requests that the authors address this hypothesis so that the evidence presented to support this hypothesis can then be used to explain the role of asymmetric action of double-bound SNF2h protomers which remains the functionally relevant state according to the authors.

Reviewer #3:

Remarks to the Author:

This is an exciting manuscript from Chio and Palovcak et al. that showcases the utility of functionalized GO grids in probing the compositional and conformational dynamics of an important biological system. The study focuses on the interactions between SNF2h and nucleosomes, and proposes a mechanism by which two copies of SNF2h act on a nucleosome in a coordinated manner to enable ATP-driven DNA translocation. A notable breakthrough in cryo-EM sample preparation that made these insights possible is the development of graphene oxide grids that had been functionalized with TAASTY polymers. Nucleosomes bound to protein cofactors have been historically recalcitrant to cryo-EM study due to interaction with the air-water interface, forcing researchers to chemically crosslink the samples or work with non-physiological buffers or high protein concentrations. There is concern that such non-physiological conditions have led to dubious mechanistic interpretations, which the authors address through the development and application of these grids to examine a previously characterized system under more physiological conditions. The functionalized GO grids protected the SNF2h-nucleosome particles from one of the disruptive air-water interfaces, preserving the more "native" interactions between nucleosomes and SNF2h and the associated conformational landscape. The cryoEM data were processed expertly and sophisticated 3D conformational analyses were used to describe distinct compositional and conformational states of the complex. Notably, these analyses enabled the authors

to ascribe trajectories of motion to the individual SNF2h moieties, identifying a previously unobserved asymmetric movement of two SNF2h subunits that gives rise to dissociation of one SNF2h from SHL6. In the context of prior biochemical, cellular, and structural studies, the findings in this paper provide important new structural insights that support a coordinated turn-based mechanism for SNF2h-mediated DNA translocation. Although the authors do not include complementary non-EM experiments to further explore or strengthen this conclusion, the work provides a critical structural framework that complements prior biophysical studies of the system that together support the proposed mechanism. This impactful study sets a new precedent for future nucleosome remodeling studies, and will be of substantial interest to structural biologists and researchers from diverse fields focused on nucleosome protein complex biology. I wholeheartedly support publication of this work in Nature Communications.

The manuscript is in great shape overall, and I laud the authors for writing a very detailed methods section. I only ask that a few additional minor details and edits be considered and addressed prior to publication:

- One of the major take-homes from the work is that the nucleosome complexes are well-protected from the air-water interface during the sample preparation process. However, there is still potential for particles to interact with the AWI in areas where the ice is too thin. There are also likely to be particles that do not adhere to the functionalized surface and interact with the AWI. The grids are referred to as "near-perfect cryo-EM grids" in the discussion, and there is a later statement:

"we find that most SNF2h-nucleosome particles that were picked could be retained for downstream processing versus the need to discard bad denatured particles using conventional Quantifoil grids."

However, 8.5 million particles were picked & extracted for processing, and less than 25% of these were used for 3D analyses – certainly not what I consider to be "most" or close to perfection. While 25% of picked particles being used for 3D is indeed a monumental improvement over traditional grids, the authors should reconsider these inflated statements. The results speak for themselves – there is no need for superlatives!

- How long do the authors anticipate these functionalized grids to remain "functional" after preparation? How soon after being prepared should they be used? Presumably they will contaminate and become more hydrophobic over time. Some mention of this would be useful in the methods.

- What type of filter paper was used for blotting and in the Vitrobot during sample preparation?

- The particles are difficult to distinguish in the images in holes with GO in Extended Data Fig 1c (particularly near the carbon edge). There are some dark smudges that might be particles, but these might be contamination. Could the authors highlight what they consider to be particles? Researchers trying to implement this methodology would benefit from this guidance.

- A specific side- and end-on views are somewhat preferred, based on the Euler distribution plots. I would prefer that the FSC plots were replaced with 3D FSC plots to convey resolution isotropy.

- A model-to-map FSC should be supplied with resolution at FSC at 0.5 denoted.

- I'm VERY surprised that a Table with data collection and modeling statistics was not included. This manuscript cannot be considered for publication without this.

I do not review anonymously, and thank the authors for publicly sharing their submitted manuscript on the bioRxiv preprint server. This practice enables others to benefit from findings presented in this research, as well as providing the authors with feedback from the community prior to completion of formal peer review.

- Gabe Lander

Reviewer #4:

Remarks to the Author:

Chio et. al. developed functionalized graphene-oxide-coated EM grids and solved the structure of non-crosslinking SNF2h-nucleosome complex at 2.3 Å. These grids protect complexes from the air-water interface and facilitate more efficient data collection. Yet, the advantages of the GO-ssDNA or GO-TAASTY grids are not characterized clearly. Additionally, whether or not these functionalized GO grids are widely suitable for other complexes is not shown. More importantly, the functional relevance of the observed conformation variability of the Snf2h motor lacks of experimental support.

1. What is the coverage and the layer number of graphene-oxide on the fabricated grids? How frequently does the partially covered GO as shown in Extended Data Fig1.c appear?
2. What are the advantages of the current functionalized GO grids compared with the graphene oxide affinity grids reported before (Wang et. al., 2020PNAS, 117(39): 24269–24273)? As a method development study, the authors should add some comparison, and include more cases of application with other protein-nucleic acid complexes.
3. Extended Data Fig3.b and c are not clear to clarify the correct register of DNA. Top views, rather than side views, of the bases probably work better.
4. The authors iterate their early controversial claim of a two-base-pair translocation for SNF2h-bound nucleosome at 140 mM KCl without ATP hydrolysis (Extended Data Fig. 6a). Because of the extensive interaction with histones within the nucleosome, the DNA translocation reaction needs to overcome a large energy barrier, and is generally believed to require the energy from ATP hydrolysis. The claim of 2 bp translocation without ATP hydrolysis is extraordinary. The current conditions allow the authors to determine the high-resolution structure with clear base identity. The reviewer encourages the authors to provide solid evidences to support their claim.
5. The authors interpret the weak density in Fig3.a right as SNF2h dissociates from the nucleosome, representing an intermediate between the ATP-bound and post-hydrolysis ADP-bound states. Alternative interpretations are that the structures are constructed from some partially broken particles, and/or unregulated breathing of the macromolecules. In cryoEM, we always see slightly different classes of particles, which may arise from the intrinsic breathing of the macromolecules (thermodynamic motion), the imperfect preparation of the sample (sample damages), measurement uncertainty, and/or different functional states. Therefore, a specular model with such an intermediate state overstretches their finding. Validation of the biological relevance of such a hypothetical intermediate state is required.
6. Likewise, the authors observed asymmetric densities of the two Snf2h protomers bound to the same nucleosome in Fig. 4, and speculate more (one protomer will hydrolyse ATP at a time) based on a speculative model above. This lacks of scientific vigor. First, the exact nucleotide states are not identified. Second, in addition to the asymmetric cases, there are complexes with symmetric or comparable densities on both sides, as shown in ED Figs. 4d and 10b.

We would like to thank all reviewers for their time and efforts providing valuable comments on our manuscript. In the following, we provide our point-to-point response to all comments.

Reviewer #1 (Remarks to the Author):

Air-water interface (AWI) has been demonstrated as a major obstacle to overcome to enhance the efficiency of cryo-EM structural determination of macromolecules. In the current work by Chio et al., the authors present a novel approach to functionalize graphene-oxide covered EM grids with specific ssDNA or TAASTY co-polymer to anchor chromatin remodeler SNF2h and nucleosome therefore preventing them from adsorbed to the AWI during cryo-EM specimen preparation. The authors reported successful preservation of complexes for high-resolution 3D cryo-EM structural determination, suggesting the effectiveness of the method's applicability for high-resolution cryo-EM of macromolecular complexes. Using the above methods, the authors successfully solved the structure of SNF2h and its complex with nucleosomes in various conformational and assembly states, providing important insights to the SNF2h's activity on nucleosomes. This work is written clearly with solid evidence to support the design and the discovery. This work will be of great interest to the field of cryo-EM and structural biology and biochemistry on protein-DNA complexes.

I recommend the work may be published with the following suggestions.

We thank the reviewer for reviewing the manuscript and for their comments and suggestions.

1. The authors should provide cryo-electron tomography analysis of the specimen to demonstrate the particle distribution in the vitreous ice. This will help understand whether the novel method has true effect to protect complexes from AWI.

We thank the reviewer for this suggestion, which indeed is good to prove that functionalized GO surfaces attract particles away from the AWI. We have now carried out cryo-electron tomography experiments with the SNF2h-nucleosome complex on ssDNA- and TAASTY-GO grids (see representative tomograms in Supplementary Movies 1 and 2 and figure below). In these representative tomograms, we clearly see an edge or fold of a GO sheet and crystalline ice contaminants, which mark the locations of both the GO surface as well as the AWI. Although the sample crowdedness and relatively small size of the SNF2h-nucleosome complex make it hard to identify every individual particles by eye, we clearly see particles in the same Z-plane as the GO and little to no particles in the Z-plane(s) with ice contamination. Therefore, in agreement with our speculation, the grids work primarily by nonspecifically adhering particles to the functionalized GO surface away from the air-water interface.

Figure A. Representative tomogram slices of a SNF2h-nucleosome sample on a (a) ssDNA-GO and (b) TAASTY-GO grid. Left images are of slices at a higher z-plane, where visible ice contaminants (red circles) mark the air-water interface. Very few to no SNF2h-nucleosome particles are observed in these slices. Right images are of slices at a lower z-plane, where GO edges (orange arrows) mark the GO surface. Many SNF2h-nucleosome particles are observed, and more visually obvious particles are marked (green circles).

We have spent quite some efforts collecting cryo-ET data of this sample, aiming to generate nicer tomograms from which we can easily see SNF2h-nucleosome particles, which are relatively small. Unfortunately, from these tomograms we collected, the particles of SNF2h-nucleosome complex are not pretty and easily seen in the slice view at a specific z-plane. They are much easier to be recognized when the tomograms are viewed as movies of slice views from one surface to another. We thus prefer only to include the movies as supplement in the manuscript, but keep the figure shown here only in the rebuttal for the reviewers.

2. The authors mentioned the strong preferential orientation of nucleosome only sample on the functionalized GO grid. Was this on ssDNA- or TAASTY- functionalized GO? The authors may elaborate more on the potential various chemical property's effect on particle orientation.

We observed strong preferred orientation with nucleosomes on ssDNA-GO grids, where nucleosome particles either sit flat on the GO surface or on their sides. We have not tried a nucleosome-alone sample on TAASTY-GO grids, but we predict the result would be similar. It is hard to provide an accurate description as to what causes the preferred orientation. Since both nucleosome and ssDNA are overall negatively charged, we can only speculate that, mediated by cations in the buffer, they interact with each other via specific regions of the nucleosome, generating preferred orientation. We now added a sentence in the text to speculate the causes of the preferred orientation (line 355).

3. The authors described a few times of "near perfect" cryo-EM grids. Only if the authors have a statistical data to show the significantly better performance on the reproducibility and successful rate of the specimen, it is dangerous to claim a "perfect" grid.

In the revised manuscript, we removed the phrase "near perfect" from lines 116, 125, and 300.

4. Fig. 3b, the two magnesium ions in the nucleotide density may be more clear with a higher thresholded density map.

We have revised the figure with a higher thresholded density map in a more clear orientation.

Reviewer #2 (Remarks to the Author):

The manuscript titled “Functionalized graphene-oxide grids enable high-resolution cryo-EM structures of the SNF2h-nucleosome complex without crosslinking” by Un Seng Chio et al. reports two new functionalized graphene-oxide support films for cryo-EM which aid in overcoming the air-water denaturation issue often encountered during cryo-EM sample preparation. The authors utilize these functionalized graphene-oxide grids to determine structures of chromatin remodeler SNF2h bound to nucleosomes in the absence of crosslinking.

In previous studies, SNF2h is suggested to function optimally as a dimer and negative stain EM structures have been able to capture two SNF2h monomers bound to nucleosomes. However, high-resolution structural characterization of the dimeric form SNF2h interacting with nucleosomes has not been successful due to limitations in sample preparation (crosslinking) and air-water denaturation issues. The authors have developed new functionalized graphene oxide grids using either ssDNA or a similar negatively charged polymer (TAASTY) to address the air-water denaturation issue. Utilizing these functionalized grids, the authors determined high-resolution structures of both single SNF2h-bound and double SNF2h-bound nucleosomes. Using particle sorting and 3D variability analysis, the authors could discern multiple states of SNF2h bound to nucleosomes for both the single-bound and double-bound configurations allowing. The authors propose a model based on these structures which explains why SNF2h functions optimally as a dimer. In summary, the authors have come up with an interesting strategy for visualizing a challenging nucleosome-bound complex. Overall, the manuscript is technically sound with an interesting new GO-based grid preparation strategy to address the air-water interface, and the authors have managed to visualize the double-bound SNF2h to nucleosome at a higher resolution compared to their previous study. However, the authors have not provided any functional assays to support the ATP hydrolysis model proposed as well as the coordinated action of the two SNF2h protomers. Therefore, it is difficult for this reviewer to relate the conformational changes visualized to the SNF2h translocation of DNA.

The main structural observations are:

1. The authors were able to visualize ADP-BeFx in the single-bound SNF2h to nucleosomes whereas in the previous structure of SNF2h bound to nucleosomes, only ADP was visualized. The authors were also able to visualize conformational changes in SNF2h with one state bound to ADP-BeFx and another with ADP bound (see related point # 3 below in comments).
2. No major conformational changes within the nucleosome in both single-bound and double-bound SNF2h. Although the authors were able to tease apart DNA unwrapping with or without the loss of density for H3, and H2A using 3D variability analysis in some states.
3. Although the authors have previously visualized double-bound SNF2h to nucleosomes at lower salt concentrations (70 mM), these reconstructions were at lower resolution $\sim 8 \text{ \AA}$. Here, the authors have managed to visualize the double-bound SNF2h to nucleosome with high-resolution structures of the SNF2h protomers. However, the authors were only able to observe ADP bound to both these protomers in the double-bound SNF2h to nucleosomes. 3D variability analysis of this double-bound SNF2h to nucleosomes suggests an interesting asymmetric conformational flexibility. However, the resolution of the protomers from the variability components precluded the unambiguous identification of nucleotide states.
4. The authors have hence inferred nucleotide states based on the engagement of the protomer with SHL+6 of the nucleosome and correlated it to the conformations observed in the single-bound SNF2h to nucleosomes. In the double-bound SNF2h structure and the corresponding

variability components, the authors infer that SNF2h stably bound to SHL+6 must have ADP-BeFx whereas SNF2h not engaged with SHL+6 has ADP.

We thank the reviewer for reviewing the manuscript and for their comments and suggestions.

Comments:

1. The authors do not directly demonstrate that the ssDNA and TAASTY functionalized GO grids retain particles away from the air-water interface. Could the authors demonstrate this using a tilt-series collection?

We thank the reviewer for this suggestion. Indeed, Reviewer 1 made the same suggestion, and we have listed our response above. Briefly, we have now performed cryo-electron tomography with the SNF2h-nucleosome complex on ssDNA- and TAASTY-GO grids (please see description above). The analysis supports that the functionalized GO grids adhere to particles bringing them away from the air-water interface.

2. The authors also do not comment on the broad use of such functionalization and whether the use of these functionalizations is restricted to nucleosome or DNA-bound protein complexes. Although this reviewer feels it might not be necessary to demonstrate their use on a protein that isn't bound to DNA or RNA, it would be helpful for the readers to understand their broad use if the authors comment on this.

We have not tested proteins and complexes not bound to DNA or RNA on these grids, so we do not have evidence to show the grids will work in these cases. However, we do believe the grids may also work for certain proteins and complexes that have charged surfaces and have revised the manuscript to include a statement to this effect (lines 365-367). In the case of RNA-protein complexes, we have not tested extensively on complexes with known structure. However, we do have data on another RNA-protein complex that is now included in a separate story, which nonetheless gave us indications that the grids can also work for RNA-protein complexes.

3. Since the nucleosome alone sample exhibited strong preferential orientation as stated by the authors, I request the authors to provide 3D FSC for all SNF2h-nucleosome maps for the readers to better assess the structures generated. In addition, could the authors comment on whether the nucleosome maps with TAASTY GO-grids also showed similar preferential orientation?

We now changed FSCs of all maps to directional FSCs (Extended Data Figs. 2b, 3d, 4a). Directional FSCs was described in our previous publication (<https://doi.org/10.1038/nature25024>). There is less preferred orientation for the SNF2h-nucleosome sample, but there are still certain views that are more dominant than others as pointed out by Reviewer 3. As discussed above (Reviewer 1 comment 2), we have not tried a nucleosome-only sample on TAASTY-GO grids, but we predict the result would be similar.

4. On page 5, line 128, the authors state that it is impossible to observe intact nucleosomes, even at concentrations of 1 μ M, without cross-linking on regular Quantifoil grids. However, this statement should perhaps be reworded as imaging of nucleosomes without crosslinking (with phase plate) has yielded high-resolution (<https://doi.org/10.1093/nar/gkw708>). However, the reviewer appreciates the intent of the original statement but requests that it be reworded to reflect the author's experience.

We intended to say that, in buffers with higher salt concentrations (e.g. 60 mM or higher), nucleosomes at 1 μ M tend to fall apart upon plunge freezing for cryo-EM on traditional Quantifoil holey carbon grids. Indeed, our experience is consistent with previous studies that nucleosomes are more stable in buffers with zero salt, and under such buffer condition cryo-EM on nucleosomes at 1 μ M without crosslinking is relatively straightforward. We have revised the sentence in the manuscript (lines 131-132).

5. It is important for the authors to provide clarification regarding the physiological salt concentration throughout the manuscript. In line 125, they assume that 150mM KCl represents a relevant concentration required for chromatin remodeling enzymes bind to the nucleosome. However, the data provided here is at 60 mM KCl and the previous studies were carried out at 140 mM KCl and 70 mM KCl. Could the conformational variability of the flanking DNA as well as the conformational flexibility observed in this study be due to the lower salt concentration overall? The authors in Extended Fig. 6 show that DNA conformational flexibility is clearly different at 140 mM KCl versus 60 mM KCl, how much influence does this flexibility have on SNF2h binding and the associated conformational changes visualized? The authors should perhaps visualize these interactions at 140 mM KCl to gauge functional relevance but since this hasn't been presented it remains difficult to correlate the observations with a functional model.

We realize that we need to clarify a few issues here. First, on line 125 (now line 128) when we mentioned 150 mM NaCl, this was mainly to make the point that at higher salt concentration, nucleosomes fall apart when freezing on traditional Quantifoil holey carbon grids as discussed above in response to the previous question.

Second, the reviewer is correct that salt concentrations can affect nucleosome dynamics. However, the effect of salt on dynamics has been shown to go in the opposite direction, where higher salt concentrations facilitate nucleosome opening. Cryo-EM data collected by the Halic lab on nucleosomes at 250 mM NaCl showed significantly more DNA unwrapping and histone H2A/H2B loss compared to data collected on nucleosomes at 50 mM NaCl (<https://doi.org/10.1038/s41594-017-0005-5>).

We hypothesize that the functionalized grids stabilized a larger of number of particles with varied conformation that would otherwise fall apart using normal holey carbon grids. The functionalized grids therefore also enabled us to observe different SNF2h conformations through 3D variability analysis that would not be observed otherwise.

6. Could the authors show a better figure showing ADP-BeFx in the variability analysis component for SNF2h bound to SHL+6? The current Fig. 3b is showing with Mg²⁺ and BeFx but it isn't clear based on the reference to Fig. 3b in text (Line 229). Perhaps a figure like that shown in Extended Figure 5b showing ADP and ADP-BeFx for this variability analysis component would be more illustrative for the readers.

We have revised the figure to be more similar to Extended Data Figure 5b.

7. One of the major conclusions from the authors that the asymmetric motion of the two SNF2h protomers are coordinated and could be related to DNA flanking length sensing and nucleotide hydrolysis is speculative as the authors do not provide any direct evidence for this conclusion. Since the authors also do not observe the DNA-length sensing HSS domains, it remains unclear how SNF2h conformation together with nucleotide state is linked to the translocation cycle.

We agree that the conclusion is speculative. Our goal was to integrate our structural data with biochemically and structurally derived models from previous studies as described below.

Based on previous single-molecule FRET studies, SNF2h has been shown to slide nucleosomes through a series of ~3 base pair nucleosome translocation steps interrupted by pause phases (<https://doi.org/10.1016/j.jmb.2022.167653>). During a pause phase the direction of nucleosome sliding can be reversed. Using covalently-connected SNF2h dimers, it has been shown that the pause phase can be lengthened if one of the two SNF2h protomers is mutated to inhibit ATPase activity, and further that this effect can be rescued by deleting the HSS of the ATPase mutant. Based on these data it has been proposed that at any given time only one of the two HSS domains can contact the corresponding flanking DNA and that ATP hydrolysis drives release of the HSS from the flanking DNA, resulting in ATP-driven sampling of DNA on either side of the nucleosome. Biochemical coupling between the HSS and the ATP state of the ATPase domain has been suggested by studies showing that the location of the HSS on a nucleosome is regulated by ATP state, such that the HSS predominantly binds flanking DNA in the apo and ADP states, and binds the nucleosome core in the ADP-BeF_x state (<https://doi.org/10.1016/j.molcel.2015.01.008>). Previous structural work on a homogenous sample of yeast ISW1 bound to nucleosomes in the ADP-bound state shows a configuration of the ATPase lobes where the contact with SHL6 is less obvious and the ATPase domain has moved upwards relative to its configuration in the presence of fully-bound ADP-BeF_x (<https://doi.org/10.1038/s41594-019-0199-9>).

Together, this collection of previous studies leads to a model where the ATPase cycle drives conformational changes in the ATPase domain that are coupled to conformational changes in the HSS that allow the two SNF2h protomers to take turns acting on a nucleosome. This model predicts that as each protomer is taking a turn, there should be structural asymmetry in how the two ATPase domains contact the nucleosome. However, such structural asymmetry had not been visualized until our study.

Our studies add to the prior working model as follows. First, the ADP-BeF_x analog is bipartite and provides the opportunity to capture multiple ATP states. This is because variations in the distance between ADP and BeF_x and in the occupancy of BeF_x can represent states along the ATP hydrolysis reaction ranging from the ATP state to the ADP state (<https://doi.org/10.1016/j.cell.2009.08.043>). In our structures determined through variability analysis we see a similar upward movement as observed previously in the ADP state that for us corresponds with missing density for BeF_x. We therefore interpreted this conformation to represent a state that is close to the ADP state (Fig. 3). Second, we identified for the first time an asymmetric state of the SNF2h dimer bound to a nucleosome as predicted by previous models. In this asymmetric state each protomer has two different conformational states, one close to the ADP state where the contact with SHL6 is released and one close to a fully bound ADP-BeF_x state, where the contact with SHL6 is maintained. To integrate these findings into the previous model, we speculate (i) that the asymmetry in the ATPase domain conformations that we observe allows the protomers to take turns and (ii) that these conformational changes are coupled to changes in the conformation of the HSS domain (despite the inability to see the HSS domain directly in our structures due to its dynamic binding nature).

Consistent with this model, as we were testing the role of the SHL6 interaction in SNF2h's proposed ATP hydrolysis cycle (see comment below), we observed that mutating the SHL6 interaction interface on SNF2h diminished the enzyme's ability to center nucleosomes (Fig. 4b). Therefore, at minimum, the new data suggests that contacts with SHL6 play an important role in proper protomer coordination.

8. The authors propose a hypothesis that SNF2h dissociation from SHL+6 is an intermediate state in the ATP hydrolysis cycle based on the conformations observed here as well as a comparison with other remodeler enzymes such as ISWI. Since the authors do not directly provide evidence for this hypothesis the conclusion hence remains a speculation in the manuscript as the authors have themselves stated. This reviewer requests that the authors address this hypothesis so that the evidence presented to support this hypothesis can then be used to explain the role of asymmetric action of double-bound SNF2h protomers which remains the functionally relevant state according to the authors.

We could not think of a technically accessible experiment to specifically test that the dissociation of the SNF2h ATPase domain from SHL6 is an intermediate state in the ATPase cycle. However, to test the functional importance of the SHL6 interaction, we generated a SNF2h mutant where we mutated all the residues we observe interacting with nucleosomal DNA at SHL6 to alanines (from ²⁹²KEKSVFKK²⁹⁹ to ²⁹²AEEAVFAA²⁹⁹), henceforth called SNF2h^{SHL6_ALA}.

First to test whether SHL6 contacts are important for SNF2h's nucleosome-stimulated ATPase activity, we performed single-turnover ATPase assays where nucleosomes are in excess of SNF2h, and SNF2h is in excess of ATP substrate (i.e. 320 nM nucleosome >> 12.5 nM SNF2h >> trace ATP). At the chosen experimental conditions, we observed a ~14-fold defect in ATP hydrolysis by SNF2h^{SHL6_ALA} compared to wild-type SNF2h (Fig. 3c). The concentration of 320 nM nucleosomes was saturating for both SNF2h^{SHL6_ALA} and SNF2h as assessed by varying nucleosome concentration. Therefore, SNF2h contacts with SHL6 play an important role in facilitating ATP hydrolysis. These data are consistent with previous observations where mutating the SHL6 interaction interface in the analogous Snf2 ATPase also resulted in decreased ATP hydrolysis activity (<https://doi.org/10.1038/nature22036>). Second, using an established gel-shift assay to monitor nucleosome remodeling by SNF2h, we observe that SNF2h^{SHL6_ALA} is still able to remodel nucleosomes, but is impaired in properly centering end-positioned nucleosomes with 60 base pairs of flanking DNA (Fig. 4b). This result is consistent with the model discussed above in response to comment #7, where alternating contacts of each SNF2h protomer with SHL6 are important for coordinating the activities of the two protomers on a nucleosome.

Further, building on the previous models described in response to comment #7, we propose that SNF2h contacts with SHL6 allosterically organize the SNF2h ATPase active site to promote ATP binding. Partial dissociation from SHL6 primes SNF2h/ISWI to hydrolyze ATP. SNF2h uses the energy from ATP hydrolysis to translocate DNA, which results in complete loss of contact with SHL6 as the ADP state is generated. To finish translocation, SNF2h remakes contact with SHL6 to exchange nucleotide, in agreement with a previously proposed model where ATP binding completes DNA translocation for Snf2 (<https://doi.org/10.1038/s41586-019-1029-2>).

We have incorporated these new results and changes into the manuscript (lines 236-246, lines 275-284, line 321, and lines 336-338).

Reviewer #3 (Remarks to the Author):

This is an exciting manuscript from Chio and Palovcak et al. that showcases the utility of functionalized GO grids in probing the compositional and conformational dynamics of an important biological system. The study focuses on the interactions between SNF2h and nucleosomes, and proposes a mechanism by which two copies of SNF2h act on a nucleosome in a coordinated manner to enable ATP-driven DNA translocation. A notable breakthrough in cryo-EM sample preparation that made these insights possible is the development of graphene oxide grids that had been functionalized with TAASTY polymers. Nucleosomes bound to protein cofactors have

been historically recalcitrant to cryo-EM study due to interaction with the air-water interface, forcing researchers to chemically crosslink the samples or work with non-physiological buffers or high protein concentrations. There is concern that such non-physiological conditions have led to dubious mechanistic interpretations, which the authors address through the development and application of these grids to examine a previously characterized system under more physiological conditions. The functionalized GO grids protected the SNF2h-nucleosome particles from one of the disruptive air-water interfaces, preserving the more “native” interactions between nucleosomes and SNF2h and the associated conformational landscape. The cryoEM data were processed expertly and sophisticated 3D conformational analyses were used to describe distinct compositional and conformational states of the complex. Notably, these analyses enabled the authors to ascribe trajectories of motion to the individual SNF2h moieties, identifying a previously unobserved asymmetric movement of two SNF2h subunits that gives rise to dissociation of one SNF2h from SHL6. In the context of prior biochemical, cellular, and structural studies, the findings in this paper provide important new structural insights that support a coordinated turn-based mechanism for SNF2h-mediated DNA translocation. Although the authors do not include complementary non-EM experiments to further explore or strengthen this conclusion, the work provides a critical structural framework that complements prior biophysical studies of the system that together support the proposed mechanism. This impactful study sets a new precedent for future nucleosome remodeling studies, and will be of substantial interest to structural biologists and researchers from diverse fields focused on nucleo-protein complex biology. I wholeheartedly support publication of this work in Nature Communications.

The manuscript is in great shape overall, and I laud the authors for writing a very detailed methods section. I only ask that a few additional minor details and edits be considered and addressed prior to publication:

We thank the reviewer for reviewing the manuscript and for their comments and suggestions.

- One of the major take-homes from the work is that the nucleosome complexes are well-protected from the air-water interface during the sample preparation process. However, there is still potential for particles to interact with the AWI in areas where the ice is too thin. There are also likely to be particles that do not adhere to the functionalized surface and interact with the AWI. The grids are referred to as “near-perfect cryo-EM grids” in the discussion, and there is a later statement:

“we find that most SNF2h-nucleosome particles that were picked could be retained for downstream processing versus the need to discard bad denatured particles using conventional Quantifoil grids.”

However, 8.5 million particles were picked & extracted for processing, and less than 25% of these were used for 3D analyses – certainly not what I consider to be “most” or close to perfection. While 25% of picked particles being used for 3D is indeed a monumental improvement over traditional grids, the authors should reconsider these inflated statements. The results speak for themselves – there is no need for superlatives!

We thank the reviewer for pointing this out. We now revised the text and removed words such as “near perfect” (lines 116, 125, and 300). We also revised “most” to “a larger percentage” (line 305). In addition, we now performed cryo-electron tomography to show that most particles are adhered to the functionalized GO surface in representative tomograms, although it is indeed still possible to have regions of the grid where the ice is too thin or to have particles that are not adhered to the GO surface as the reviewer points out. We should also clarify that the 8.5 million particles picked by cryoSPARC’s template picker still contained a lot of “junk” that are not

necessarily nucleosomes (e.g. GO edges and folds, carbon/carbon edges, ice contaminants, etc.), so the actual percentage of picked nucleosomal particles that were retained for 3D processing is likely much higher. However, we have not conducted a thorough quantitative analysis to provide an exact number here.

- How long do the authors anticipate these functionalized grids to remain “functional” after preparation? How soon after being prepared should they be used? Presumably they will contaminate and become more hydrophobic over time. Some mention of this would be useful in the methods.

We have not thoroughly tested the shelf life of the grids after preparation. We typically have used the grids within a few days after preparation. We have added this detail in the methods (lines 623-624).

- What type of filter paper was used for blotting and in the Vitrobot during sample preparation?

We used Whatman 1 qualitative filter papers for all blotting and have now added this detail in the methods (lines 637, 640).

- The particles are difficult to distinguish in the images in holes with GO in Extended Data Fig 1c (particularly near the carbon edge). There are some dark smudges that might be particles, but these might be contamination. Could the authors highlight what they consider to be particles? Researchers trying to implement this methodology would benefit from this guidance.

Due to the crowdedness of the sample, it is indeed difficult to pick out individual particles by eye. We have now circled representative particles in each image.

- A specific side- and end-on views are somewhat preferred, based on the Euler distribution plots. I would prefer that the FSC plots were replaced with 3D FSC plots to convey resolution isotropy.

We have now replaced the FSC plots with directional FSC plots. The calculation of directional FSC plots was described in our previous publication (<https://doi.org/10.1038/nature25024>).

- A model-to-map FSC should be supplied with resolution at FSC at 0.5 denoted.

We have now included model-to-map FSC's for all models built (Extended Data Fig. 9e).

- I'm VERY surprised that a Table with data collection and modeling statistics was not included. This manuscript cannot be considered for publication without this.

We sincerely apologize for this oversight! We have now included the information in a table.

I do not review anonymously, and thank the authors for publicly sharing their submitted manuscript on the bioRxiv preprint server. This practice enables others to benefit from findings presented in this research, as well as providing the authors with feedback from the community prior to completion of formal peer review.

- Gabe Lander

Reviewer #4 (Remarks to the Author):

Chio et. al. developed functionalized graphene-oxide-coated EM grids and solved the structure of non-crosslinking SNF2h-nucleosome complex at 2.3 Å. These grids protect complexes from the air-water interface and facilitate more efficient data collection.

Yet, the advantages of the GO-ssDNA or GO-TAASTY grids are not characterized clearly.

With respect, we disagree with this point. We demonstrated that the ssDNA- and TAASTY-GO grids work well for determining high-resolution structures of the SNF2h-nucleosome complex without the need for chemical crosslinking, and with additional advantages including lower amounts of sample needed for grid preparation and higher particle retention during data processing.

Additionally, whether or not these functionalized GO grids are widely suitable for other complexes is not shown.

We do not claim ssDNA-GO or TAASTY-GO grids are widely suitable for many other complexes. In our hands, we only have the SNF2h-nucleosome complex as a model test case that consistently falls apart during grids preparation without using cross-linking. Since the functionalized grids work very well for this sample, it is an alternative method that is worth trying if anyone encounters a similar problem in other samples. We did try this approach successfully in preparing a sample of an RNA-protein complex, which is now included in a separate manuscript.

More importantly, the functional relevance of the observed conformation variability of the Snf2h motor lacks of experimental support.

As discussed above in response to Reviewer 2 comments 7 and 8 and described in more detail below in response to comments 5 and 6, we have now performed experiments to address the functional importance of the contacts we see between SNF2h and nucleosomal DNA at SHL6, and the asymmetric contacts we see with our doubly-bound data.

1. What is the coverage and the layer number of graphene-oxide on the fabricated grids? How frequently does the partially covered GO as shown in Extended Data Fig1.c appear?

We have previously described a simplified GO layering procedure on Quantifoil grids (<https://doi.org/10.1016/j.jsb.2018.07.007>) that we also used in this study. We estimate that >90% of the grid surface is covered with GO, with ~40% monolayer, 40% bilayer, and <20% with three or more layers. The partially covered GO image as shown in Extended Data Fig. 1c is rare and was shown only for illustrative purposes.

2. What are the advantages of the current functionalized GO grids compared with the graphene oxide affinity grids reported before (Wang et. al., 2020PNAS, 117(39): 24269–24273)? As a method development study, the authors should add some comparison, and include more cases of application with other protein-nucleic acid complexes.

The GO functionalization reported in Wang et al. 2020 described a specific type of affinity grids that work through covalent interaction between SpyTag and SpyCatcher. Unless we engineer either SpyTag or SpyCatcher to one of the protein components, this approach would not work for the SNF2h-nucleosome complex. We did not try this specific affinity approach to avoid extensive and time-consuming experiments to test appropriate tagging positions that do not perturb protein functions. The ssDNA-GO and TAASTY-GO grids work through non-specific affinity and are suited for our targeted SNF2h-nucleosome complex. As discussed above, we have also used the

ssDNA- and TAASTY-GO grids on an RNA-protein complex, but that data will be published as part of a separate story.

3. Extended Data Fig3.b and c are not clear to clarify the correct register of DNA. Top views, rather than side views, of the bases probably work better.

We appreciate this point, and included now top views of the base pair indicated with an asterisk that best illustrates that one orientation is wrong versus the other.

4. The authors iterate their early controversial claim of a two-base-pair translocation for SNF2h-bound nucleosome at 140 mM KCl without ATP hydrolysis (Extended Data Fig. 6a). Because of the extensive interaction with histones within the nucleosome, the DNA translocation reaction needs to overcome a large energy barrier, and is generally believed to require the energy from ATP hydrolysis. The claim of 2 bp translocation without ATP hydrolysis is extraordinary. The current conditions allow the authors to determine the high-resolution structure with clear base identity. The reviewer encourages the authors to provide solid evidences to support their claim.

We believe this is beyond the scope of the current study. Here we are only comparing the structure determined in this study at 60 mM KCl with the previous highest resolution structure determined at 140 mM KCl and stating that we do not see the 2 base pair translocation at 60 mM KCl. In addition, an explanation for the 2 base pair translocation at 140 mM KCl is already provided in the previous study, as well as data from both ensemble and single-molecule FRET assays that corroborate the observation of a 2 base pair translocation at 140 mM KCl but not at 70 mM KCl.

5. The authors interpret the weak density in Fig3.a right as SNF2h dissociates from the nucleosome, representing an intermediate between the ATP-bound and post-hydrolysis ADP-bound states. Alternative interpretations are that the structures are constructed from some partially broken particles, and/or unregulated breathing of the macromolecules. In cryoEM, we always see slightly different classes of particles, which may arise from the intrinsic breathing of the macromolecules (thermodynamic motion), the imperfect preparation of the sample (sample damages), measurement uncertainty, and/or different functional states. Therefore, a specular model with such an intermediate state overstretches their finding. Validation of the biological relevance of such a hypothetical intermediate state is required.

We have now tested the mechanistic relevance of the proposed intermediate. Specifically, we generated a SNF2h mutant where we mutated all the residues interacting with nucleosomal DNA at SHL6 to alanines (from ²⁹²KEKSVFKK²⁹⁹ to ²⁹²AEEAVFAA²⁹⁹) called SNF2h^{SHL6_ALA}. SNF2h^{SHL6_ALA} is over 10-fold defective in ATP hydrolysis relative to wild-type SNF2h under single-turnover conditions (Fig. 3c; see above response to Reviewer 2 comment 8 for discussion). These results are consistent with the model we proposed, where SHL6 contacts help promote ATP binding to the SNF2h ATPase active site. In this model, partial dissociation from SHL6 primes SNF2h/ISWI to hydrolyze ATP. SNF2h uses the energy from ATP hydrolysis to translocate DNA, which results in complete loss of contact with SHL6 as the ADP state is generated. To finish translocation, SNF2h remakes contact with SHL6 to exchange nucleotide, in agreement with a previously proposed model where ATP binding completes DNA translocation for Snf2 (<https://doi.org/10.1038/s41586-019-1029-2>).

Further we note that we are not proposing the entirety of the model based solely on data in this study. Our goal is to integrate our findings with models that have been previously proposed based on substantial biochemical and structural studies as described above in response to Reviewer 2 comment 7. We also summarize these previous models below.

Based on previous single-molecule FRET studies, SNF2h has been shown to slide nucleosomes through a series of ~3 base pair nucleosome translocation steps interrupted by pause phases (<https://doi.org/10.1016/j.jmb.2022.167653>). During a pause phase the direction of nucleosome sliding can be reversed. Using covalently-connected SNF2h dimers, it has been shown that the pause phase can be lengthened if one of the two SNF2h protomers is mutated to inhibit ATPase activity and further that this effect can be rescued by deleting the HSS of the ATPase mutant. Based on these data it has been proposed that at any given time only one of the two HSS domains can contact the corresponding flanking DNA and that ATP hydrolysis drives release of the HSS from the flanking DNA, resulting in ATP-driven sampling of DNA on either side of the nucleosome. Biochemical coupling between the HSS and the ATP state of the ATPase domain has been suggested by studies showing that the location of the HSS on a nucleosome is regulated by ATP state, such that the HSS predominantly binds flanking DNA in the apo and ADP states and binds the nucleosome core in the ADP-BeF_x state (<https://doi.org/10.1016/j.molcel.2015.01.008>). Previous structural work on a homogenous sample of yeast ISW1 bound to nucleosomes in the ADP-bound state shows a configuration of the ATPase lobes where the contact with SHL6 is less obvious and the ATPase domain has moved upwards relative to its configuration in the presence of fully-bound ADP-BeF_x (<https://doi.org/10.1038/s41594-019-0199-9>). Together this collection of previous studies leads to a model where the ATPase cycle drives conformational changes in the ATPase domain that are coupled to conformational changes in the HSS that allow the two SNF2h protomers to take turns acting on a nucleosome. This model predicts that as each protomer is taking a turn, there should be structural asymmetry in how the two ATPase domains contact the nucleosome. However, such structural asymmetry had not been visualized until our study.

6. Likewise, the authors observed asymmetric densities of the two Snf2h protomers bound to the same nucleosome in Fig. 4, and speculate more (one protomer will hydrolyse ATP at a time) based on a speculative model above. This lacks of scientific vigor. First, the exact nucleotide states are not identified. Second, in addition to the asymmetric cases, there are complexes with symmetric or comparable densities on both sides, as shown in ED Figs. 4d and 10b.

As discussed in response to comment #5, our new structural work extends and strengthens previous models, which were based on substantial biochemical and structural work. Specifically, our studies add to the previous model as follows. First, the ADP-BeF_x analog is bipartite and provides the opportunity to capture multiple ATP states. This is because variations in the distance between ADP and BeF_x and in the occupancy of BeF_x can represent states along the ATP hydrolysis reaction ranging from the ATP state to the ADP state (<https://doi.org/10.1016/j.cell.2009.08.04>). In our structures of singly-bound SNF2h determined through variability analyses we can clearly identify the nucleotide state. In these structures we see a similar upward movement as observed previously in the ADP state (<https://doi.org/10.1038/s41594-019-0199-9>) that for us corresponds with missing density for BeF_x. We therefore interpreted this conformation to represent a state that is close to the ADP state (Fig. 3). Second, we identified for the first time an asymmetric state of the SNF2h dimer bound to a nucleosome as predicted by previous models. Here, as the reviewer points out we indeed cannot determine the exact nucleotide state directly from the structures, and we are extrapolating the observations made with the higher-resolution singly-bound structures. In the asymmetric state each protomer has two different conformational states, one close to the ADP state where the contact with SHL6 is released and one close to a fully bound ADP-BeF_x state, where the contact with SHL6 is maintained. To integrate these findings into the previous model, we speculate (i) that the asymmetry in the ATPase domain conformations that we observe, allows the protomers to take turns and (ii) that these conformational changes are coupled to changes in

the conformation of the HSS domain (despite the inability to see the HSS domain directly in our structures due to its dynamic binding nature).

Our new data with SNF2h^{SHL6_ALA} showing that the SHL6 binding mutant is defective in properly centering end-positioned nucleosomes suggests that the asymmetric contacts we see with SHL6 are indeed important for proper coordination between the two SNF2h protomers. Our model does not explicitly exclude the existence of complexes with comparable SNF2h densities on both sides. Such complexes may exist if both SNF2h protomers are bound with ATP but the HSS domain has not yet changed conformation to drive hydrolysis (<https://doi.org/10.1016/j.molcel.2015.01.008>), and this is depicted within the length-sensing state in our model (Fig. 4). The ability of ADP-BeF_x to adopt a range of ATP states likely make these states also accessible for visualization.

Reviewers' Comments:

Reviewer #1:

Remarks to the Author:

All of my concerns have been addressed in the revised manuscript.

Reviewer #2:

Remarks to the Author:

The authors have provided satisfactory responses to all of my comments and the comments of other reviewers. I thank the authors for a detailed explanation of my comments. As such, I am happy with the revised manuscript and the authors' additions.

Reviewer #3:

Remarks to the Author:

The authors have addressed my concerns, and I have no further issues to raise. I believe the manuscript can proceed to publication.

-gabe

Reviewer #4:

Remarks to the Author:

The reviewer appreciates the idea of making modified GO grids, which are potentially very useful for the analyses of nucleosome-bound complexes. Yet, the conclusions on the SNF2h remodeling mechanism, including the importance of SHL6 release and the coordination of the two protomers bound on the same nucleosome, are less convincing. Some important statements of the revised manuscript need to be clarified.

To validate the proposed model, the authors included new biochemical assays, the ATPase and remodeling activities, in the revised manuscript. These are highly appreciated. Yet, the mutant design seems to provide little or weak support for their hypothesis that the SNF2h dissociation from SHL6 represents an intermediate state, entering a state primed for ATP hydrolysis. The loss of activity of the SNF2hSHL6_ALA mutant showed in the revised manuscript supports the importance of SHL6 binding, but not the importance of SHL6 dissociation. A more convincing way to validate their model is to make mutants with the opposite strategy: enhancing SHL6 binding and preventing SHL6 release.

The defect in nucleosome centering even after completion of the reaction (Fig. 4b) may be due to enzyme inactivation after some time under the experimental conditions, but not related to the coordination between SNF2h protomers. Clarification of this point is important.

In Fig. 3b, the authors show that only ADP is clearly visible. Because of the poor EM density in this state, it is also possible that the large errors in the structure alignment, and/or heterogeneity of the structures averaged out the BeFx signal, which is intrinsically weaker than the protein signals. So, BeFx may be there, but skips the detection because of the overall poor EM density. Clarification of this point is crucial.

Double-bound structures are also observed in other chromatin remodelers, including Snf2 and ACL1. The Snf2 and ALC1 motors do not seem to function as a dimer. Can the authors comment on these structures?

Reviewer #1 (Remarks to the Author):

All of my concerns have been addressed in the revised manuscript.

We thank reviewer 1, for their helpful comments and are happy their concerns have been addressed.

Reviewer #2 (Remarks to the Author):

The authors have provided satisfactory responses to all of my comments and the comments of other reviewers. I thank the authors for a detailed explanation of my comments. As such, I am happy with the revised manuscript and the authors' additions.

We thank reviewer 2 for their helpful comments and are glad they are happy with the revised manuscript.

Reviewer #3 (Remarks to the Author):

The authors have addressed my concerns, and I have no further issues to raise. I believe the manuscript can proceed to publication.

-gabe

We thank reviewer 3 for their helpful comments and are happy to know reviewer 3 feels the manuscript can go to publication.

Reviewer #4 (Remarks to the Author):

The reviewer appreciates the idea of making modified GO grids, which are potentially very useful for the analyses of nucleosome-bound complexes. Yet, the conclusions on the SNF2h remodeling mechanism, including the importance of SHL6 release and the coordination of the two protomers bound on the same nucleosome, are less convincing. Some important statements of the revised manuscript need to be clarified.

We thank the reviewer for their thoughtful suggestions for testing the mechanistic model. Below we address the specific concerns raised by the reviewer about the mechanistic conclusions.

To validate the proposed model, the authors included new biochemical assays, the ATPase and remodeling activities, in the revised manuscript. These are highly appreciated. Yet, the mutant design seems to provide little or weak support for their hypothesis that the SNF2h dissociation from SHL6 represents an intermediate state, entering a state primed for ATP hydrolysis. The loss of activity of the SNF2h^{SHL6_ALA} mutant showed in the revised manuscript supports the importance of SHL6 binding, but not the importance of SHL6 dissociation. A more convincing way to validate their model is to make mutants with the opposite strategy: enhancing SHL6 binding and preventing SHL6 release.

We agree that the SNF2h^{SHL6_ALA} mutant does not directly demonstrate the putative role of SNF2h dissociation from SHL6. We also appreciate the reviewer's suggestion on generating mutants that enhance SHL6 binding, but this would require developing an assay to verify that the mutants are specifically engaging SHL6 more strongly, as well as screening various mutations near the interaction interface since most of the interacting residues are already positively charged. We feel that these open-ended explorations are beyond the scope of the current study. So instead we decided to test the importance of the contact more generally by disrupting the putative contact. Our results are consistent with a model where the contact made by SNF2h at SHL6 is important for the remodeling cycle. We have added a sentence to clarify this point in the discussion of the main text (lines 333-335).

The defect in nucleosome centering even after completion of the reaction (Fig. 4b) may be due to enzyme inactivation after some time under the experimental conditions, but not related to the coordination between SNF2h protomers. Clarification of this point is important.

We do not believe the observed defect is due to enzyme inactivation after some time because the mutant is still able to hydrolyze ATP at the same rate in the same assay buffer and temperature after at least 3 to 4 hours as shown by the time courses over this time-scale (time courses now shown in new Supplementary Figure 11). It is also worth noting that a SpyTag-SpyCatcher connected SNF2h dimer with one monomer defective in ATP hydrolysis is still able to center nucleosomes properly although at a slower rate (Leonard JD & Narlikar GJ, Mol. Cell 2015), suggesting the nucleosome centering defect we see with SNF2h^{SHL6_ALA} cannot be fully explained as due to slower ATP hydrolysis.

In Fig. 3b, the authors show that only ADP is clearly visible. Because of the poor EM density in this state, it is also possible that the large errors in the structure alignment, and/or heterogeneity of the structures averaged out the BeFx signal, which is intrinsically weaker than the protein signals. So, BeFx may be there, but skips the detection because of the overall poor EM density. Clarification of this point is crucial.

We appreciate the reviewer pointing this out, and we should indeed clarify that the EM density in state 2B (i.e. the SHL6-dissociated state) is poor *relative* to the EM density in state 2A. At lower threshold levels, the SNF2h density appears much better than as depicted in Fig. 3a (see Figure below). Based on local resolution estimates from cryoSPARC, the map for state 2B still has a local resolution between 2.5 to 3 Å in the region for nucleotide, which should be sufficient to visualize BeFx if it is present, but we do not see such density. We have edited the sentence in the manuscript to "Dissociation of SNF2h from SHL+6 leads to overall weaker density for SNF2h relative to SNF2h density when bound..." to help clarify this point.

astructure 2B map at different contour levels**b**structure 2B map local resolutionstructure 2B nucleotide local resolution
Double-bound structures are also observed in other chromatin remodelers, including Snf2 and ACL1. The Snf2 and ALC1 motors do not seem to function as a dimer. Can the authors comment on these structures?

We appreciate this comment, and to our knowledge, we do not believe there is data that explicitly rules out the possibility that the Snf2 and ALC1 motors alone cannot bind and function as dimers on nucleosomes at sufficiently high concentrations. However, unlike for SNF2h, there is no data to suggest the two protomers coordinate their remodeling in the context of Snf2 and ALC1. Specifically for SNF2h, it was shown that the rate constant for SNF2h-mediated nucleosome remodeling is cooperative with a Hill coefficient of 1.8 (Racki LR et al., Nature 2009), and a SpyTag-SpyCatcher connected SNF2h dimer can remodel nucleosomes faster than SNF2h monomers under conditions where enzyme concentration is lower than nucleosome concentration (Leonard JD & Narlikar GJ, Mol. Cell 2015). In addition, *in vivo*, Snf2 is part of the >1MDa SWI/SNF complex that contains many other subunits that likely preclude two SWI/SNF complexes from engaging the same nucleosome substrate. For ALC1, the optimal substrate is a PARylated nucleosome (Bacic et al., eLife 2021), and such a nucleosome might only be modified on one face. Therefore, one ALC1 protomer would have an advantage in binding versus a second protomer. A previous study on the ALC1-nucleosome complex indeed presents a kinetic model that allows for the possibility of a second ALC1 engaging the nucleosome and remodeling at a slower rate (please see Figure 5 figure supplement 1 from Bacic et al., eLife 2021). We have now included a paragraph discussing these two remodelers in the discussion section of the manuscript.